# Linear programming-based stabilization and synchronization of positive complex networks with dynamic link subsystems

**Shouting Hong**[1], **Junfeng Zhang** [1]*, **Gang Zheng**[2], **Haoyue Yang**[1], **Bhatti Uzair Aslam**[1]

**1** School of Information and Communication Engineering, Hainan University, Haikou, China, **2** Inria, University of Lille, CNRS, Centrale Lille, Lille, France

* jfzhang@hainanu.edu.cn

**Data availability statement:** All relevant data are within the paper and its Supporting Information files.

**Funding:** This research was supported by the National Natural Science Foundation of China (62463007, 62463005, and 62073111), Natural Science Foundation of Hainan Province (625RC710 and 625MS047), Science Research Funding of Hainan University (KYQD(ZR)22180), and Postgraduate Innovative Research Funding of Hainan Province (Qhys2023-278, Qhys2023-279 and Qhys2023-280).

## Abstract

This paper investigates the stabilization and synchronization of positive complex networks with dynamic links. A class of positive complex networks is constructed by introducing dynamic link between nodes. A controller and the corresponding coupling term with respect to links are designed to achieve the positivity and stability of complex networks and link systems, respectively. Then, a synchronization controller and the corresponding coupling term with respect to links and node states are proposed, respectively. The main contributions are as follows: (i) A novel coupling term is constructed to achieve stability and synchronization of positive complex networks, (ii) A stabilization and synchronization framework is constructed by designing two classes of controllers and coupling terms, and (iii) A tractable design, analysis, and computation method is introduced by virtue of linear programming and copositive Lyapunov function. Finally, a simulation example is provided to verify the effectiveness and feasibility of the proposed approaches.

## Introduction

Complex networks (CNs) have become the focus in the field of control theory and engineering due to their high interconnection, nonlinearity, and time-varying nature. It covers a wide range of complex systems from power systems to ecological and transportation networks [1]. In practice, there is a complex class of systems whose dynamic behavior maintains non-negativity under non-negative initial conditions. Such systems are defined as positive systems [2]. In the past few decades, positive system theory has received extensive attention due to its applicability in various fields such as economics, communications, and biology, etc. [3–5]. Due to the unique advantages of positive systems, the combination of CNs and positive systems forms positive complex networks (PCNs). This allows for good characterization and optimization of real-world systems with complex characteristics and non-negative constraints. In transportation networks, CNs formed by traffic flow can be modeled using PCNs because the traffic flow keeps non-negative. Since the power output and input are non-negative, PCNs can also be used for modeling power systems. Therefore, PCNs have potential

**Competing interests:** The authors have declared that no competing interests exist.

applications in accurately capturing and analyzing real-world systems with positive dynamics and interactions.

The current research on PCNs remains in an early stage. Multiple challenges arise when combining positive systems with complex networks. First, there are essential differences in research approaches between positive and non-positive systems. The design and analysis of positive systems often require specific strategies such as co-positive Lyapunov function (CLF) and linear programming (LP) [6–8]. In addition, traditional complex networks do not consider positive constraints and the positivity of PCNs increases the complexity of the design. At present, there is a lack of a unified and efficient design framework to integrate positive system theory with complex networks. Therefore, a comprehensive framework needs to be constructed to smooth the study of PCNs. This includes not only the development of new theories and methods, but also the exploration of synchronization strategies that can effectively deal with positive constraints.

Synchronization of complex networks has attracted much attention and various synchronization frameworks have been established such as pinning synchronization [9], exponential synchronization [10], local synchronization [11], etc. [12,13]. To effectively address the modeling and control of complex nonlinear systems, T-S fuzzy systems have emerged as a significant tool. T-S fuzzy systems characterize the behavior of the systems using a set of "IF-THEN" fuzzy rules, each corresponding to a linear subsystem [14]. This approach breaks down complex nonlinear issues into multiple local linear problems, thus the modeling and control processes are simplified. Therefore, the combination of synchronization framework and fuzzy systems can more effectively analyze and solve synchronization problems in CNs. This approach simplifies the modeling and control process by decomposing a nonlinear problem into multiple local linear problems. The cluster synchronization of fuzzy CNs with semi-Markovian switching characteristics was studied using the inequality analysis technique [15]. A fuzzy memory pinning impulsive control was proposed to solve the pinning synchronization problem of stochastic fuzzy time-delayed CNs [16]. The bipartite synchronization of signed networks with time-varying delays was solved based on fuzzy systems [17]. These studies demonstrate the wide applications and effectiveness of fuzzy systems in the synchronization problems of CNs.

In fact, the topology of the network changes in CNs due to random changes such as node failures, environmental disturbances, and machine failures. These changes can be characterized by the topological state described by the Markovian chain. Markovian jump models can effectively capture system mutations caused by random environmental disturbances and node interconnection changes. Therefore, Markovian jump systems (MJSs) are widely used to model these uncertainties and mutations in practical engineering applications. Positive MJSs (PMJSs) refer to a type of MJSs with positive characteristics [18]. So far, a large number of research results have been accumulated in the field of PMJSs. A nonlinear CLF was used to design a non-fragile saturation control of nonlinear PMJSs [19]. The event-triggered state feedback and dynamic output feedback control for PMJSs have been proposed [20]. The double sensitive fault detection filter for PMJSs is explored [21]. In the field of Markovian jump complex networks (MJCNs), synchronization is a core problem. A number of relevant research results have been reported [22–25]. Several criteria for stochastic synchronization of MJCNs were derived by constructing time-dependent Lyapunov-Krasovskii functions and applying inequality techniques [22]. Based on Lyapunov's stability theory, a sample data controller was proposed to ensure the robust synchronization of CNs [23]. A passive synchronization criterion was established for MJCNs by combining integral inequality, free weighting matrix, and convex combination methods [24]. The finite-time synchronization problem of MJCNs was studied based on dissipative theory and a new delay-dependent finite-time

stability dissipation rule was derived in the form of linear matrix inequality (LMI) [25]. The above research mainly focuses on the synchronization of MJCNs and the stochastic synthesis of PMJSs. Despite these remarkable advances, the synchronization of fuzzy positive MJCNs (PMJCNs) remains challenging.

In most results of CNs, the links between nodes are set as a fixed connection manner. Although many remarkable achievements have been made in the analysis and synthesis of nodal dynamics [26–28], the change of links should not be ignored in CNs, especially during the transmission of data, information, and material. This involves energy distribution in power systems, cargo transportation in logistics networks, and signal transmission in biological networks. By precisely controlling the state of each link, the overall performance of the network can be significantly improved [29–31]. Some important progress has been made in the study of the dynamics of positive link systems. The consensus of links in node networks was discussed [32–34], where the links of the networks were described by positive systems. The non-negative link consensus of network systems was analyzed using state feedback protocols [35]. Although the dynamic characteristics of nodes and links have been considered for CNs [36,37], the dynamic coupling mechanism of links has not been discussed for PCNs. This paper attempts to fill this gap in the field. In addition, real systems are often affected by unpredictable internal and external factors and perturbations can affect the stability and synchronization of the system. Therefore, it is necessary to consider the perturbation in PCNs.

Inspired by the above results, this paper studies the stabilization and synchronization problems of PCNs coupled by node subsystems (NSs) and link subsystems (LSs). The main contributions of this paper are as follows: (i) A model of fuzzy PMJCNs composed of NSs and LSs coupling is proposed, (ii) A novel coupling term is designed in LSs to achieve stability and synchronization of the NSs, (iii) The $L_1$-gain controller is designed to achieve the stochastic stability and synchronization of fuzzy PMJCNs with $L_1$-gain performance, and (iv) A simple design and analysis approach is presented using LP and CLF. The paper is structured as follows: Some preliminaries are provided in the section of Problem formulation, the section of Main results presents the stabilization and synchronization of PCNs, an example is given in the section of Illustrative example, and the main conclusions of this paper are summarized in the section of Conclusions.

**Notation** The symbols $\mathbb{R}^n$ and $\mathbb{R}^{n \times n}$ denote the set of $n$-dimensional real vectors and the set of $n \times n$ real matrices, respectively. $\mathcal{N}^+$ represents the set of positive integers. The Kronecker product of two matrices $A$ and $B$ is represented as $A \otimes B$. The identity matrix in $\mathbb{R}^n$ is denoted by $I_n$. For a matrix $A$, the notation $A^\top$ signifies its transpose and $a_{ij}$ represents the entry at the $i$th row and $j$th column. Additionally, $A \succ 0 \ (\succeq 0)$ means that all elements $a_{ij}$ are non-negative. A matrix is considered as Metzler if the off-diagonal elements are non-negative. Define $\mathbf{1}_n = (\underbrace{1, \dots, 1}_{n})^\top$, and the $n$-dimensional vector with a single entry of one at the $t$th position is denoted by $\mathbf{1}_n^{(t)} = (\underbrace{0, \dots, 0}_{t-1}, 1, \underbrace{0, \dots, 0}_{n-t})^\top$. The summation $\sum_{b,n,r=1}^{m}$ is denoted as:

$\sum_{b=1}^{m} \sum_{n=1}^{m} \sum_{r=1}^{m}$. 1-norm of a vector $x$ is defined by $\|x\|_1$. Given a function $\omega(t): [0, \infty) \to \mathbb{R}^n$, its $L_1$ norm is defined as: $\|\omega(t)\|_{L_1} = \int_0^\infty \|\omega(t)\|_1 \, \mathrm{d}t$. Denote by the $L_1$ space $L_1([0, \infty), \mathbb{R}^n) \triangleq \{\omega(t) \mid \|\omega(t)\|_{L_1} < \infty\}$.

## Problem formulation

Consider fuzzy PMJCNs with the controlled node $i$ described as:

**Rule** $b$: IF $\psi_1(t)$ is $\phi_{b1}$, $\psi_2(t)$ is $\phi_{b2}$, $\cdots$, and $\psi_g(t)$ is $\phi_{bg}$, THEN

$$
\begin{aligned}
\dot{x}_i(t) \quad &= A_b(r_t)x_i(t) + F_b(r_t)f(x_i(t)) + \kappa \sum_{j=1}^{N} l_{ij}(t)\Lambda_b(r_t)h_j(x_j(t)) \\
&\quad + B_b(r_t)u_i(t) + E_b(r_t)\omega_i(t), \\
y_i(t) \quad &= C_b(r_t)x_i(t), \quad i = 1, 2, \cdots, N,
\end{aligned}
\tag{1}
$$

where $x_i(t) = (x_{i1}(t), x_{i2}(t), \ldots, x_{in}(t))^\top \in \mathbb{R}^n$, $y_i(t) \in \mathbb{R}^n$, $u_i(t) \in \mathbb{R}^n$ denote state, output and control input of node $i$, respectively; $l_{ij}(t) \in \mathbb{R}$ is the time-varying weighted value of the link from node $j$ to node $i$; $w_i(t) \in \mathbb{R}^n$ represents the disturbance that satisfies $E\{\int_0^\infty \|\omega(t)\|_1\}dt < \overline{\omega}$ and $\overline{\omega} > 0$ is given; $f(x_i(t)) = (f_1(x_{i1}(t)), \ldots, f_n(x_{in}(t)))^\top \in \mathbb{R}^n$ denotes a vector-valued nonlinear function and $h_j(x_j(t)) = (h_{j1}(x_{j1}(t)), \ldots, h_{jn}(x_{jn}(t)))^\top \in \mathbb{R}^n$ is a coupled function; $\Lambda_p(r_t)$ represents the jumping inner coupling configuration matrices of the system which is a positive diagonal matrix; $\kappa > 0$ is the coupling strength, $A_b(r_t)$ is a Metzler matrix, $B_b(r_t) \succeq 0$, $C_b(r_t) \succeq 0$, $F_b(r_t) \succeq 0$, and $E_b(r_t) \succeq 0$; $\psi(t) = (\psi_1(t), \psi_2(t), \ldots, \psi_g(t))^\top$ are the premise variables; $\phi_{b1}, \phi_{b2}, \ldots, \phi_{bg}$ are fuzzy sets; $b$ is the number of IF-THEN rules and $b = 1, \cdots, m$ denotes the $b$th rule; $r_t$ indicates a continuous-time Markovian process and it represents the system mode taking numbers from a limited set $\mathfrak{S} = \{1, 2, \ldots, J\}$, $J \in \mathcal{N}^+$. Moreover, the transition probabilities are given by

$$
Pr(r_{t+\Delta} = q | r_t = p) = \begin{cases} \lambda_{pq}\Delta + o(\Delta), & p \neq q, \\ 1 + \lambda_{pp}\Delta + o(\Delta), & p = q, \end{cases}
$$

where $p, q \in \mathfrak{S}$, $\Delta t > 0$, and $\lim_{\Delta t \to 0} \frac{o(\Delta t)}{\Delta t} = 0$; $\lambda_{pq} \geq 0 (p \neq q)$ is the transition rate of the mode jump from mode $p$ to mode $q$ when time goes from $t$ to $t + \Delta t$ and satisfies $\lambda_{pp} = -\sum_{q \in \mathfrak{S}, q \neq p} \lambda_{pq}$. By the fuzzy inference method, the defuzzified fuzzy system (1) is rewritten as:

$$
\begin{aligned}
\dot{x}_i(t) \quad &= \sum_{b=1}^{m} \Theta_b(\phi(t))(A_b(r_t)x_i(t) + F_b(r_t)f(x_i(t)) \\
&\quad + \kappa \sum_{j=1}^{N} l_{ij}(t)\Lambda_b(r_t)h_j(x_j(t)) + B_b(r_t)u_i(t) + E_b(r_t)w_i(t)), \\
y_i(t) \quad &= \sum_{b=1}^{m} \Theta_b(\phi(t))(C_b(r_t)x_i(t)),
\end{aligned}
\tag{2}
$$

where $\Theta_b(\phi(t))$ is the membership function and $\Theta_b(\phi(t)) = \dfrac{\Pi_{j=1}^{g}\phi_{bj}(\psi_b(t))}{\sum_{b=1}^{m} \Pi_{j=1}^{g}\phi_{bj}(\psi_b(t))} \geq 0$, $\sum_{b=1}^{m} \Theta_b(\phi(t)) = 1$.

**Assumption 1** The nonlinear functions $f(x_i(t))$ and $h_i(x_i(t))$ have a slope restriction:

$$
\varepsilon \leq \frac{f_k(x_{ik}(t)) - f_k(s(t))}{x_{ik}(t) - s_k(t)} \leq \epsilon, \rho \leq \frac{h_{ik}(x_{ik}(t)) - h_{ik}(s(t))}{x_{ik}(t) - s_k(t)} \leq \varrho,
\tag{3}
$$

where $x_{ik}(t)$ represents the $k$th element of $x_i(t)$ for $k \in \{1, 2, \ldots, n\}$, $0 < \varepsilon < \epsilon$, $0 < \rho < \varrho$, $f_k(0) = 0$, and $h_{ik}(0) = 0$.

**Remark 1** Assumption 1 sets the slope constraint for the nonlinear part of the system. Such an assumption has also been applied to nonlinear problems [38–40]. Multiplying all terms of the slope inequality by $(x_{ik}(t) - s_k(t))^2 > 0$ and choosing $s_k(t) = 0$, the slope restriction condition is transformed into a sector inequality. The sector condition ensures that the response of the nonlinearity is kept in the first and third quadrants under given parameter conditions [41–43]. Therefore, the slope restriction is more stringent than the sector condition. Moreover, Assumption 1 can smooth the positivity of the systems.

**Assumption 2** For the inner coupling function $h_i(x_i(t))$, it holds that $0 \leq h_{ik}(x_i(t)) \leq \bar{h}$ for all $i = 1, 2, \cdots, N$ and $k = 1, 2, \cdots, n$.

**Remark 2** Generally, a norm bounded condition is usually used for CNs [44]. This assumption reflects the practical limitation in describing finite energy. The condition $h_{ik}(x_i(t)) \geq 0$ is also frequently encountered in network control issues. Thus, each element of the inner coupling function $h_i(x_i(t))$ is bounded by 0 and $\bar{h}$ for PCNs.

Next, some preliminaries on positive systems are introduced.

**Definition 1** [2] A system is positive if for any non-negative initial conditions, inputs, and disturbances, all states and outputs are non-negative.

**Definition 2** [45] The system (1) is stochastically $L_1$ stable for the initial state $x_0 \in \mathbb{R}^n$ and mode $r_0 \in \mathfrak{S}$ if the following conditions are satisfied:

(i) When $\omega(t) = 0$, the system is stochastically stable, that is,

$$\mathbb{E}\left\{\int_0^\infty \|x(t)\|_1 dt | x_0, r_0\right\} < \infty.$$

(ii) There exists a positive constant $\gamma$ such that

$$\mathbb{E}\left\{\int_0^\infty \|y(t)\|_1 dt | x_0, r_0\right\} < \gamma \mathbb{E}\left\{\int_0^\infty \|\omega(t)\|_1 dt\right\}$$

holds for zero initial state and $\omega(t) \in L_1([0, \infty), \mathbb{R}^n)$.

**Lemma 1** [2] A matrix $A$ is Metzler if and only if there exists a constant $\alpha$ such that $A + \alpha I \succeq 0$.

**Lemma 2** [2] The system

$$\dot{x}(t) = Ax(t) + Bu(t) + E\omega(t),$$
$$y(t) = Cx(t),$$

is positive if and only if $A$ is Metzler, $B \succeq 0$, $C \succeq 0$, and $E \succeq 0$.

## Main results

This section consists of two sections. First, the positivity and stochastic $L_1$ stability of NSs and LSs are addressed by designing two different coupling terms and the corresponding controllers. Then, the synchronization of fuzzy PMJCNs is presented.

### Stabilization

Assume that the stability of LSs is only affected by the topology between links and is not related to NSs. It means that the coupling term design of LSs is independent of the node state. For sake of simplicity, set $r_t = p$. Based on the fuzzy rule, the controller $u_i(t)$ is designed as:

**Rule $n$**: IF $\psi_1(t)$ is $\phi_{n1}$, $\psi_2(t)$ is $\phi_{n2}$, $\cdots$, and $\psi_g(t)$ is $\phi_{ng}$, THEN

$$u_i(t) = K_{np}x_i(t),$$

where $K_{np}$ is the control gain matrix to be determined. Then, the defuzzified controller is rewritten as:

$$u_i(t) = \sum_{n=1}^m \Theta_n(\phi(t))K_{np}x_i(t). \tag{4}$$

Define $X(t) = (x_1^\top(t), x_2^\top(t), \ldots, x_N^\top(t))^\top \in \mathbb{R}^{nN}$, $F(x(t)) = (f^\top(x_1(t)), f^\top(x_2(t)), \ldots, f^\top(x_N(t)))^\top \in \mathbb{R}^{nN}$, $H(x(t)) = (h_1(x_1(t)), h_2(x_2(t)), \ldots, h_N(x_N(t))) \in \mathbb{R}^{n \times N}$, and $W(t) =$

$(\omega_1^\top(t), \omega_2^\top(t), ..., \omega_N^\top(t))^\top \in \mathbb{R}^{nN}$. By (1) and (4), and the properties of Kronecker product, the following closed-loop system is obtained:

$$
\begin{aligned}
\dot{X}(t) &= \sum_{b,n=1}^{m} \Theta_b(\phi(t))\Theta_n(\phi(t))\big((I_N \otimes (A_{bp} + B_{bp}K_{np}))X(t) + (I_N \otimes F_{bp})F(x(t)) \\
&\quad + \kappa(I_N \otimes \Lambda_{bp}H(x(t)))L(t) + (I_N \otimes E_{bp})W(t)\big), \\
Y(t) &= \sum_{b=1}^{m} \Theta_b(\phi(t))\big((I_N \otimes C_{bp})X(t)\big).
\end{aligned}
\tag{5}
$$

The model of LSs can be expressed as:

**Rule** $b$: IF $\psi_1(t)$ is $\phi_{b1}$, $\psi_2(t)$ is $\phi_{b2}$, $\cdots$, and $\psi_g(t)$ is $\phi_{bg}$, THEN

$$
\dot{L}_i(t) = Q_{bp}L_i(t) + \Gamma_i(t),
\tag{6}
$$

where $L_i(t) = (l_{i1}(t), l_{i2}(t), ..., l_{iN}(t))^\top \in \mathbb{R}^N$ is the state vector of incoming links of node $i$, $Q_{bp} \in \mathbb{R}^{N \times N}$ is given matrix, and $\Gamma_i(t) \in \mathbb{R}^N$ is the coupling term. Using the properties of the Kronecker product, the system (6) is defuzzified into:

$$
\dot{L}(t) = \sum_{b=1}^{m} \Theta_b(\phi(t))\big((I_N \otimes Q_{bp})L(t) + \Gamma(t)\big),
\tag{7}
$$

where $L(t) = (L_1^\top(t), L_2^\top(t), ..., L_N^\top(t))^\top \in \mathbb{R}^{N^2}$, $\Gamma(t) = (\Gamma_1^\top(t), \Gamma_2^\top(t), ..., \Gamma_N^\top(t))^\top \in \mathbb{R}^{N^2}$, and the coupling term $\Gamma_i(t)$ is designed using fuzzy model:

**Rule** $r$: IF $\psi_1(t)$ is $\phi_{b1}$, $\psi_2(t)$ is $\phi_{b2}$, $\cdots$, and $\psi_g(t)$ is $\phi_{bg}$, THEN

$$
\Gamma_i(t) = K_{rp}L_i(t),
$$

where $K_{rp}$ is the control gain matrix to be determined for LSs. Then, the defuzzified coupling term is rewritten as:

$$
\Gamma_i(t) = \sum_{r=1}^{m} \Theta_r(\phi(t))\big(K_{rp}L_i(t)\big).
\tag{8}
$$

**Theorem 1** If there exist constants $\underline{\beta} > 0$, $\overline{\beta} > 0$, $\underline{\gamma} > 0$, $\overline{\gamma} > 0$, $\kappa > 0$, $\mu_0 > 0$, $\mu_1 > 0$, $\mu_2 > 1$, $\mu_3 > 0$, $\mathbb{R}^n$ vectors $z_{np}^{(\iota)} \prec 0$, $\nu_p \succ 0$, $\nu_q \succ 0$, $\mathbb{R}^N$ vectors $z_{rp}^{(\iota)} \succ 0$, $\tau_p \succ 0$, $\tau_q \succ 0$ such that

$$
A_{bp} + B_{bp}\sum_{\iota=1}^{n} \mathbf{1}_n^{(\iota)} z_{np}^{(\iota)\top} + \varepsilon F_{bp} + \mu_0 I_n \succeq 0,
\tag{9a}
$$

$$
\underline{\beta} \leq \mathbf{1}_m^{(\iota)\top} \tau_p \leq \overline{\beta},
\tag{9b}
$$

$$
Q_{bp} + \frac{1}{\underline{\beta}}\sum_{\iota=1}^{N} \mathbf{1}_m^{(\iota)} z_{rp}^{(\iota)\top} + \mu_1 I_N \succeq 0,
\tag{9c}
$$

$$
\underline{\gamma} \leq \mathbf{1}_n^{(\iota)\top} B_{bp}^\top \nu_p \leq \overline{\gamma},
\tag{9d}
$$

$$
(A_{bp}^\top + \varepsilon F_{bp}^\top)\nu_p + \underline{\gamma}\sum_{\iota=1}^{n} z_{np}^{(\iota)} + \sum_{q=1}^{J} \lambda_{pq}\nu_q + \mu_2 C_{bp}^\top \mathbf{1}_n \prec 0,
\tag{9e}
$$

$$
Q_{bp}^\top \tau_p + \sum_{\iota=1}^{N} z_{rp}^{(\iota)} + \kappa H^\top \Lambda_{bp}^\top \nu_p + \sum_{q=1}^{J} \lambda_{pq}\tau_q + \mu_3 \mathbf{1}_N \prec 0,
\tag{9f}
$$

$$
E_{bp}^\top \nu_p - \gamma \mathbf{1}_n \prec 0,
\tag{9g}
$$

hold for each $p, q \in \mathfrak{S}$, then the systems (5) and (7) are positive and stochastically $L_1$ stable under the reliable controller (4) with

$$K_{np} = \sum_{\iota=1}^{n} \mathbf{1}_n^{(\iota)} z_{np}^{(\iota)\top}, \tag{10}$$

and the coupling term (8) with

$$K_{rp} = \sum_{\iota=1}^{N} \frac{\mathbf{1}_m^{(\iota)} z_{rp}^{(\iota)\top}}{\mathbf{1}_m^{(\iota)\top} \tau_p}. \tag{11}$$

**Proof:** First, the positivity of the system (7) is considered. From the coupling term (8), the system (7) can be rewritten as:

$$\dot{L}(t) = \sum_{b,r=1}^{m} \Theta_b(\phi(t))\Theta_r(\phi(t))\left((I_N \otimes (Q_{bp} + K_{rp}))L(t)\right). \tag{12}$$

From (9b), (9c), and (11), we have $Q_{bp} + K_{rp} + \mu_1 I_N \succeq 0$. Thus, $Q_{bp} + K_{rp}$ is Metzler by Lemma 1 and the system (7) is positive by Lemma 2. Then, the positivity of the system (5) is achieved. Using Assumption 1, it is easy to have

$$\dot{X}(t_0^+) \succeq \sum_{b,n=1}^{m} \Theta_b(\phi(t_0))\Theta_n(\phi(t_0))(\mathcal{A}X(t_0) + \mathcal{B}W(t_0)),$$

where

$$\mathcal{A} = \begin{pmatrix} \aleph_{11} & \kappa\rho l_{12}(t_0)\Lambda_{bp} & \cdots & \kappa\rho l_{1N}(t_0)\Lambda_{bp} \\ \kappa\rho l_{21}(t_0)\Lambda_{bp} & \aleph_{22} & \cdots & \kappa\rho l_{2N}(t_0)\Lambda_{bp} \\ \vdots & \vdots & \ddots & \vdots \\ \kappa\rho l_{N1}(t_0)\Lambda_{bp} & \kappa\rho l_{N2}(t_0)\Lambda_{bp} & \cdots & \aleph_{NN} \end{pmatrix},$$

$\aleph_{ii} = A_{bp} + B_{bp}K_{np} + \varepsilon F_{bp} + \rho\kappa l_{ii}(t_0)\Lambda_{bp}$ and $\mathcal{B} = I_N \otimes E_{bp}$. From (9a) and (10), it derives that $A_{bp} + B_{bp}K_{np} + \varepsilon F_{bp} + \mu_0 I_n \succeq 0$. By Lemma 1, $A_{bp} + B_{bp}K_{np} + \varepsilon F_{bp}$ is Metzler, that is, $A_{bp} + B_{bp}K_{np} + \varepsilon F_{bp} + \kappa\rho l_{ii}(t_0)\Lambda_{bp}$ is Metzler for $i = \{1, 2, \dots, N\}$. Noting the condition $l_{ij}(t_0) > 0$, it yields that $\kappa\rho l_{ij}(t_0)\Lambda_{bp} \succeq 0$ for $i \neq j$. Then, $\mathcal{A}$ is Metzler. Since $E_{bp} \succeq 0$, then $\mathcal{B} \succeq 0$. Thus, we have $\dot{X}(t_0^+) \succeq 0$ for $\dot{X}(t_0) \succeq 0$. Then, it is easy to give that $X(t) \succeq 0$ for any initial state $X(t_0) \succeq 0$ using recursive derivation. Therefore, the system (5) is positive by Definition 1.

Consider the following candidate CLF:

$$V(X_c(t)) = X^\top(t)(\mathbf{1}_N \otimes \nu) + L^\top(t)(\mathbf{1}_N \otimes \tau).$$

Then, the weak infinitesimal operator of $V(X_c(t))$ is

$$\begin{aligned} \Gamma V(X_c(t)) &= \sum_{b,n,r=1}^{m} \Theta_b(\phi(t))\Theta_n(\phi(t))\Theta_r(\phi(t)) \\ &\times \left( X^\top(t)(\mathbf{1}_N \otimes (A_{bp}^\top + K_{np}^\top B_{bp}^\top)\nu_p \right. \\ &+ \sum_{q=1}^{J} \lambda_{pq}(\mathbf{1}_N \otimes \nu_q)) + F^\top(x(t))(\mathbf{1}_N \otimes F_{bp}^\top \nu_p) \\ &+ L^\top(t)(\mathbf{1}_N \otimes \kappa H^\top \Lambda_{bp}^\top \nu_p + \mathbf{1}_N \otimes (Q_{bp}^\top + K_{rp}^\top)\tau_p \\ &\left. + \sum_{q=1}^{J} \lambda_{pq}(\mathbf{1}_N \otimes \tau_q)) + W^\top(t)(\mathbf{1}_N \otimes E_{bp}^\top \nu_p) \right). \end{aligned} \tag{13}$$

From Assumptions 1 and 2, we have

$$F^\top(x(t))(\mathbf{1}_N \otimes \nu_p) \le \epsilon X^\top(t)(\mathbf{1}_N \otimes \nu_p),$$
$$\kappa L^\top(t)(\mathbf{1}_N \otimes H^\top(x(t))\Lambda_{bp}^\top \nu_p \le \kappa L^\top(t)(\mathbf{1}_N \otimes H^\top)\Lambda_{bp}^\top \nu_p,$$

where $H = \begin{pmatrix} \bar{h}_{11} & \bar{h}_{12} & \cdots & \bar{h}_{1N} \\ \bar{h}_{21} & \bar{h}_{22} & \cdots & \bar{h}_{2N} \\ \vdots & \vdots & \ddots & \vdots \\ \bar{h}_{n1} & \bar{h}_{n2} & \cdots & \bar{h}_{nN} \end{pmatrix}$.

Then, it deduces that

$$
\begin{aligned}
\Gamma V(X_c(t), r_t = p) \quad &\le \sum_{b,n,r=1}^m \Theta_b(\phi(t))\Theta_n(\phi(t))\Theta_r(\phi(t))(X^\top(t)(\mathbf{1}_N \\
&\otimes (A_{bp}^\top + K_{np}^\top B_{bp}^\top)\nu_p + \mathbf{1}_N \otimes \epsilon F_{bp}^\top \nu_p + \sum_{q=1}^J \lambda_{pq}(\mathbf{1}_N \otimes \nu_q)) \\
&+ L^\top(t)(\mathbf{1}_N \otimes \kappa H^\top \Lambda_{bp}^\top \nu_p + \mathbf{1}_N \otimes (Q_{bp}^\top + K_{rp}^\top)\tau_p \\
&+ \sum_{q=1}^J \lambda_{pq}(\mathbf{1}_N \otimes \tau_q)) + W^\top(t)(\mathbf{1}_N \otimes E_{bp}^\top \nu_p)).
\end{aligned}
\tag{14}
$$

By (9d), (10), and (11), it gives that

$$
\begin{aligned}
\Gamma V(X_c(t), r_t) \quad &\le \sum_{b,n,r=1}^m \Theta_b(\phi(t))\Theta_n(\phi(t))\Theta_r(\phi(t))(X^\top(t)(\mathbf{1}_N \\
&\otimes (A_{bp}^\top \nu_p + \underline{\gamma} \sum_{l=1}^n z_{np}^{(l)}) + \epsilon(\mathbf{1}_N \otimes F_{bp}^\top \nu_p) + \sum_{q=1}^J \lambda_{pq}(\mathbf{1}_N \otimes \nu_q)) \\
&+ W^\top(t)(\mathbf{1}_N \otimes E_{bp}^\top \nu_p) + L^\top(t)(\mathbf{1}_N \otimes (Q_{bp}^\top \tau_p + \sum_{l=1}^l z_{rp}^{(l)}) \\
&+ \kappa(\mathbf{1}_N \otimes H^\top \Lambda_{bp}^\top \nu_p) + \sum_{q=1}^J \lambda_{pq}(\mathbf{1}_N \otimes \tau_q))).
\end{aligned}
$$

Consider the case $W(t) = 0$. Combining (9e) and (9f) yields that

$$
\begin{aligned}
\Gamma V(X_c(t), r_t) \quad &\le \sum_{b=1}^m \Theta_b(\phi(t))(-\mu_2 X^\top(t)(\mathbf{1}_N \otimes C_{bp}^\top \mathbf{1}_n) \\
&- \mu_3 L^\top(t)(\mathbf{1}_N \otimes \mathbf{1}_N)) \\
&\le \sum_{b=1}^m \Theta_b(\phi(t))(-\mu_2 \vartheta_b \|X\|_1 - \mu_3 \|L\|_1),
\end{aligned}
\tag{15}
$$

where $\vartheta_b = \min_{p \in \mathfrak{S}}\{\min_{i=1,2,\dots,n} \Sigma_j^n C_{bp}^{(ij)}\}$ and $C_{bp}^{(ij)}$ is the $i$th row $j$th column element of $C_{bp}$. Applying Dynkin's formula to (15) gives

$$
\begin{aligned}
&\mathbb{E}\{V(X_c(t), r_t)\} - V(X_c(0), r_0) \\
&\le \sum_{b=1}^m \Theta_b(\phi(t))(-\mu_2 \vartheta_b \mathbb{E}\{\int_0^t \|X(s)\|_1 ds | X_c(0), r_0\} - \mu_3 \mathbb{E}\{\int_0^t \|L(s)\|_1 ds | L_0, r_0\}).
\end{aligned}
\tag{16}
$$

Due to $\mathbb{E}\{V(x(t), r_t = p)\} > 0$, we can obtain

$$\lim_{t\to\infty} \mathbb{E}\{\int_0^t \|X(s)\|_1 ds | X_c(0), r_0\} \le \sum_{b=1}^m \Theta_b(\phi(t)) \frac{1}{\mu_2 \vartheta_b} V(X_c(0), r_0) < \infty,$$

$$\lim_{t\to\infty} \mathbb{E}\{\int_0^t \|L(s)\|_1 ds | L_0, r_0\} \le \frac{1}{\mu_3} V(X_c(0), r_0) < \infty.$$

Therefore, the systems (5) and (7) are stochastically stable by Definition 2.

Next, consider the case $W(t) \neq 0$. Under the zero initial condition, we have $\mathbb{E}\{V(X_c(t), r_t)\} = \mathbb{E}\{\int_0^t \Gamma V(X_c(t), r_t)\} > 0$. Then,

$$
\begin{aligned}
\mathbb{E}\left\{\int_0^t (\|Y(t)\|_1 - \gamma \|W(t)\|_1) dt\right\} \leq{} & \sum_{b,n,r=1}^m \Theta_b(\phi(t)) \Theta_n(\phi(t)) \Theta_r(\phi(t)) (\mathbb{E}\{\int_0^t (X^\top(t) \\
& \times (I_N \otimes C_{bp}^\top) \mathbf{1}_{nN} - \gamma W^\top(t) \mathbf{1}_{nN} + X^\top(t) (\mathbf{1}_N \\
& \otimes (A_{bp}^\top \nu_p + \underline{\gamma} \sum_{t=1}^n z_{np}^{(t)}) + \epsilon(\mathbf{1}_N \otimes F_{bp}^\top \nu_p) \\
& + \sum_{q=1}^J \lambda_{pq} (\mathbf{1}_N \otimes \nu_q)) + W^\top(t) (\mathbf{1}_N \otimes E_{bp}^\top \nu_p) \\
& + L^\top(t) (\mathbf{1}_N \otimes (Q_{bp}^\top \tau_p + \sum_{t=1}^l z_{rp}^{(t)}) \\
& + \kappa (\mathbf{1}_N \otimes H^\top \Lambda_{bp}^\top \nu_p) + \sum_{q=1}^J \lambda_{pq} (\mathbf{1}_N \otimes \tau_q))) dt\}).
\end{aligned}
\tag{17}
$$

By (9e) and (9f), it gives that

$$
\begin{aligned}
\mathbf{1}_N \otimes ((A_{bp}^\top + \epsilon F_{bp}^\top) \nu_p + \underline{\gamma} \sum_{t=1}^n z_{np}^{(t)} + \sum_{q=1}^J \lambda_{pq} \nu_q) &\prec \mathbf{1}_N \otimes (-\mu_2 C_{bp}^\top \mathbf{1}_n), \\
\mathbf{1}_N \otimes (Q_{bp}^\top \tau_p + \sum_{t=1}^N z_{rp}^{(t)} + \kappa H^\top \Lambda_{bp}^\top \nu_p + \sum_{q=1}^J \lambda_{pq} \tau_q) &\prec \mathbf{1}_N \otimes (-\mu_3 \mathbf{1}_N).
\end{aligned}
$$

Then,

$$
\begin{aligned}
\mathbb{E}\left\{\int_0^t (\|Y(t)\|_1 - \gamma \|W(t)\|_1) dt\right\} \leq{} & \sum_{b=1}^m \Theta_b(\phi(t)) (\mathbb{E}\{\int_0^t (1 - \mu_2) X^\top(t) (I_N \\
& \otimes C_{bp}^\top) \mathbf{1}_{nN} - \mu_3 L^\top(t) (\mathbf{1}_N \otimes \mathbf{1}_N) \\
& + W^\top(t) (\mathbf{1}_N \otimes (E_{bp}^\top \nu_p - \gamma \mathbf{1}_n)) dt\}).
\end{aligned}
\tag{18}
$$

Together with (9g) gives $\mathbb{E}\{\int_0^\infty (\|Y(t)\|_1 - \gamma \|W(t)\|_1) dt\} < 0$. By Definition 2, the systems (5) and (7) are stochastically $L_1$ stable. □

**Remark 3** The analysis and synthesis of nodal dynamics were addressed in [26–28]. It is assumed that the coefficient of the coupling term is fixed. Indeed, the coefficient may change owing to the change of node dynamics, environmental change, and other unexpected factors. Consequently, the change will affect the stability of NSs. It is interesting to design a dynamical coupling term and explore how the links affect the behavior of the node. In Theorem 1, a controller and a coupling term are designed such that the node and the link dynamic are simultaneously positive and stable.

**Remark 4** How to define the positivity of a system is key to investigate positive systems. Up to now, there is no unified framework on the positivity definition of various systems. Although there have been many results on CNs [12–16], few results are devoted to PCNs. Existing results on positive systems cannot be applied for PCNs [18–21]. Theorem 1 presents a design approach to the positivity of PCNs by designing a controller (4) and a coupling term (8). It should be noted that the coupling design of LSs is only related to the state of the link. A further consideration is that the coupling design is related to the state of the node.

In Theorem 1, the coupling design of LSs is independent of the node state. Theorem 2 will discuss the stochastic stability of fuzzy PMJCNs by designing a more general coupling term, that is, the coupling term is dependent of the node state.

Based on the fuzzy model, the LSs are expressed as follows:

**Rule** $b$: IF $\psi_1(t)$ is $\phi_{b1}$, $\psi_2(t)$ is $\phi_{b2}$, $\cdots$, and $\psi_g(t)$ is $\phi_{bg}$, THEN

$$
\dot{L}_i(t) = Q_{bp} L_i(t) + \Gamma_i(x(t)),
\tag{19}
$$

where $x(t) = (x_1^\top(t), x_2^\top(t), \ldots, x_N^\top(t))^\top \in \mathbb{R}^{nN}$ and $\Gamma_i(x(t)) \in \mathbb{R}^N$ is the coupling term for the state of the nodes. Define $L(t) = (L_1^\top(t), L_2^\top(t), \ldots, L_N^\top(t))^\top \in R^{N^2}$ and $\Gamma(x(t)) =$

$(\Gamma_1^\top(x(t)), \Gamma_2^\top(x(t)), \dots, \Gamma_N^\top(x(t)))^\top \in R^{N^2}$. Using the properties of the Kronecker product, the system (19) is defuzzified into:

$$\dot{L}(t) = \sum_{b=1}^m \Theta_b(\phi(t))\big((I_N \otimes Q_{bp})L(t) + \Gamma(x(t))\big). \tag{20}$$

Then, the coupling term of LSs in relation to the state of the node is designed as:

$$\Gamma(x(t)) = -(I_N \otimes (\xi_p \mathbf{1}_N^\top + \kappa M^\top \Lambda_{bp} H(x(t))))L(t), \tag{21}$$

where $M \succ 0$ with $M \in \mathbb{R}^{n \times N}$ is a given matrix.

**Theorem 2** If there exist constants $\underline{\alpha} > 0, \overline{\alpha} > 0, \kappa > 0, \mu_0 > 0, \mu_1 > 0, \mu_2 > 1, \mu_3 > 0, \gamma > 0, \mathbb{R}^n$ vectors $z_{np}^{(t)} < 0, \eta_p > 0, \eta_q > 0$, and $\mathbb{R}^N$ vectors $\xi_p > 0, \xi_q > 0$ such that

$$Q_{bp} - \xi_p \mathbf{1}_N^\top - \kappa M^\top \Lambda_{bp} H + \mu_0 I_N \succeq 0, \tag{22a}$$

$$A_{bp} + B_{bp} \sum_{t=1}^n \mathbf{1}_n^{(t)} z_{np}^{(t)\top} + \varepsilon F_{bp} + \mu_1 I_n \succeq 0, \tag{22b}$$

$$\underline{\alpha} \le \mathbf{1}_n^{(t)\top} B_{bp}^\top \eta_p \le \overline{\alpha}, \tag{22c}$$

$$A_{bp}^\top \eta_p + \underline{\alpha} \sum_{t=1}^n z_{np}^{(t)} + \varepsilon F_{bp}^\top \eta_p + \sum_{q=1}^J \lambda_{pq} \eta_q + \mu_2 C_{bp}^\top \mathbf{1}_n < 0, \tag{22d}$$

$$Q_{bp}^\top \xi_p + \sum_{q=1}^J \lambda_{pq} \xi_q + \mu_3 \mathbf{1}_N < 0, \tag{22e}$$

$$E_{bp}^\top \eta_p - \gamma \mathbf{1}_n < 0, \tag{22f}$$

hold for each $p, q \in \mathfrak{S}$, then the systems (5) and (20) are positive and stochastically $L_1$ stable under the reliable controller (4) with (10) and the coupling term (21).

**Proof:** Give the initial state $L(t_0) \succeq 0$. From the coupling term (21), the system (20) can be rewritten at time $t_0$ as:

$$\dot{L}(t_0^+) = \sum_{b=1}^m \Theta_b(\phi(t_0))(I_N \otimes (Q_{bp} - \xi_p \mathbf{1}_N^\top - \kappa M^\top \Lambda_{bp} H(x(t_0))))L(t_0). \tag{23}$$

From Assumption 2, it follows that $-\kappa M^\top \Lambda_{bp} H(x(t_0)) \succeq -\kappa M^\top \Lambda_{bp} H$. Then, the system (23) becomes:

$$\dot{L}(t_0^+) \succeq \sum_{b=1}^m \Theta_b(\phi(t_0))(I_N \otimes (Q_{bp} - \xi_p \mathbf{1}_N^\top - \kappa M^\top \Lambda_{bp} H))L(t_0). \tag{24}$$

From (22a) and Lemma 1, it is easy to have that $Q_{bp} - \xi_p \mathbf{1}_N^\top - \kappa M^\top \Lambda_{bp} H$ is Metzler. Thus, it follows that $\dot{L}(t_0^+) \succeq 0$ given $\dot{L}(t_0) \succeq 0$. Then, it can be readily shown that $L(t) \succeq 0$ for any initial state $L(t_0) \succeq 0$ through recursive derivation. Therefore, the system (20) is positive by Lemma 2.

Choose the stochastic CLF: $V(X_c(t), r_t = p) = X_c^\top(t)\varsigma$, where $X_c(t) = (X^\top(t), L^\top(t))^\top$, $\varsigma = (\mathbf{1}_N^\top \otimes \eta_p^\top, \mathbf{1}_N^\top \otimes \xi_p^\top)^\top$, and $\eta_p = M\xi_p$. Then, the weak infinitesimal operator of $V(X_c(t), r_t)$ is given as:

$$\begin{aligned}
\Gamma V(X_c(t), r_t) = \sum_{b,n=1}^m \Theta_b(\phi(t))\Theta_n(\phi(t))(X^\top(t)(\mathbf{1}_N \otimes (A_{bp}^\top + K_{np}^\top B_{bp}^\top)\eta_p) \\
- \kappa L^\top(t)(\mathbf{1}_N \otimes H^\top(x(t))\Lambda_{bp}^\top M\xi_p) + \kappa L^\top(t)(\mathbf{1}_N \otimes H^\top(x(t))\Lambda_{bp}^\top \eta_p) \\
+ W^\top(t)(\mathbf{1}_N \otimes E_{bp}^\top \eta_p) + X^\top(t)\sum_{q=1}^J \lambda_{pq}(\mathbf{1}_N \otimes \eta_q)
\end{aligned}$$

$$+ L^{\top}(t)(\mathbf{1}_N \otimes Q_{bp}^{\top}\xi_p) + F^{\top}(x(t))(\mathbf{1}_N \otimes F_{bp}^{\top}\eta_p)$$

$$- L^{\top}(t)(\mathbf{1}_N \otimes \mathbf{1}_N \xi_p^{\top}\xi_p - \sum_{q=1}^{J} \lambda_{pq}(\mathbf{1}_N \otimes \xi_q))). \tag{25}$$

Since $\eta_p = M\xi_p$, it yields that

$$\begin{aligned}
\Gamma V(X_c(t), r_t) \quad &= \sum_{b,n=1}^{m} \Theta_b(\phi(t))\Theta_n(\phi(t))(X^{\top}(t)(\mathbf{1}_N \otimes (A_{bp}^{\top} + K_{np}^{\top}B_{bp}^{\top})\eta_p) \\
&+ F^{\top}(x(t))(\mathbf{1}_N \otimes F_{bp}^{\top}\eta_p) + \kappa L^{\top}(t)(\mathbf{1}_N \otimes H^{\top}(x(t))\Lambda_{bp}^{\top}M\xi_p) \\
&+ W^{\top}(t)(\mathbf{1}_N \otimes E_{bp}^{\top}\eta_p) + X^{\top}(t)\sum_{q=1}^{J}\lambda_{pq}(\mathbf{1}_N \otimes \eta_q) \\
&+ L^{\top}(t)(\mathbf{1}_N \otimes Q_{bp}^{\top}\xi_p) + L^{\top}(t)(\sum_{q=1}^{J}\lambda_{pq}(\mathbf{1}_N \otimes \xi_q) \\
&- \mathbf{1}_N \otimes (\kappa H^{\top}(x(t))\Lambda_{bp}^{\top}M\xi_p + \mathbf{1}_N\xi_p^{\top}\xi_p))).
\end{aligned} \tag{26}$$

From Assumption 1, we have that $F^{\top}(x(t)) \preceq \epsilon X^{\top}(t)$. Then,

$$\begin{aligned}
\Gamma V(X_c(t), r_t) \quad &\leq \sum_{b,n=1}^{m} \Theta_b(\phi(t))\Theta_n(\phi(t))(X^{\top}(t)(\mathbf{1}_N \otimes (A_{bp}^{\top} + K_{np}^{\top}B_{bp}^{\top})\eta_p) \\
&+ \epsilon X^{\top}(t)(\mathbf{1}_N \otimes F_{bp}^{\top}\eta_p) + W^{\top}(t)(\mathbf{1}_N \otimes E_{bp}^{\top}\eta_p) \\
&+ X^{\top}(t)\sum_{q=1}^{J}\lambda_{pq}(\mathbf{1}_N \otimes \eta_q) + L^{\top}(t)(\mathbf{1}_N \otimes Q_{bp}^{\top}\xi_p) \\
&- L^{\top}(t)(\mathbf{1}_N \otimes \mathbf{1}_N\xi_p^{\top}\xi_p) + L^{\top}(t)\sum_{q=1}^{J}\lambda_{pq}(\mathbf{1}_N \otimes \xi_q)).
\end{aligned} \tag{27}$$

By (22c) and (10), it gives

$$\begin{aligned}
\Gamma V(X_c(t), r_t) \quad &\leq \sum_{b,n=1}^{m} \Theta_b(\phi(t))\Theta_n(\phi(t))(X^{\top}(t)(\mathbf{1}_N \otimes (A_{bp}^{\top}\eta_p + \underline{\alpha}\sum_{\iota=1}^{n} z_{np}^{(\iota)}) \\
&+ \epsilon(\mathbf{1}_N \otimes F_{bp}^{\top}\eta_p) + \sum_{q=1}^{J}\lambda_{pq}(\mathbf{1}_N \otimes \eta_q)) + W^{\top}(t)(\mathbf{1}_N \otimes E_{bp}^{\top}\eta_p) \\
&+ L^{\top}(t)(\mathbf{1}_N \otimes (Q_{bp}^{\top}\xi_p - \mathbf{1}_N\xi_p^{\top}\xi_p) + \sum_{q=1}^{J}\lambda_{pq}(\mathbf{1}_N \otimes \xi_q))).
\end{aligned}$$

Since $\xi_p \succeq 0$, then $-\mathbf{1}_N\xi_p^{\top}\xi_p \prec 0$. Therefore,

$$\begin{aligned}
\Gamma V(X_c(t), r_t) \quad &\leq \sum_{b,n=1}^{m} \Theta_b(\phi(t))\Theta_n(\phi(t))(X^{\top}(t)(\mathbf{1}_N \otimes (A_{bp}^{\top}\eta_p + \underline{\alpha}\sum_{\iota=1}^{n} z_{np}^{(\iota)}) \\
&+ \epsilon(\mathbf{1}_N \otimes F_{bp}^{\top}\eta_p) + \sum_{q=1}^{J}\lambda_{pq}(\mathbf{1}_N \otimes \eta_q)) + W^{\top}(t)(\mathbf{1}_N \otimes E_{bp}^{\top}\eta_p) \\
&+ L^{\top}(t)(\mathbf{1}_N \otimes Q_{bp}^{\top}\xi_p + \sum_{q=1}^{J}\lambda_{pq}(\mathbf{1}_N \otimes \xi_q))).
\end{aligned}$$

Consider the case $W(t) = 0$. Combining (22d) and (22e) yields that

$$\begin{aligned}
\mathbf{1}_N \otimes (A_{bp}^{\top}\eta_p + \underline{\alpha}\sum_{\iota=1}^{n} z_{np}^{(\iota)} + \epsilon(\mathbf{1}_N \otimes F_{bp}^{\top}\eta_p)) &\prec \mathbf{1}_N \otimes (-\mu_2 C_{bp}^{\top}\mathbf{1}_n), \\
\mathbf{1}_N \otimes Q_{bp}^{\top}\xi_p + \sum_{q=1}^{J}\lambda_{pq}(\mathbf{1}_N \otimes \xi_q) &\prec \mathbf{1}_N \otimes (-\mu_3\mathbf{1}_N).
\end{aligned}$$

Then,

$$\Gamma V(X_c(t), r_t) \leq \sum_{b=1}^{m} \Theta_b(\phi(t))(-\mu_2\vartheta_b\|X\|_1 - \mu_3\|L\|_1). \tag{28}$$

Applying Dynkin's formula to (28) gives

$$\begin{aligned}
&\mathbb{E}\{V(X_c(t), r_t)\} - V(X_c(0), r_0) \\
&\leq \sum_{b=1}^{m} \Theta_b(\phi(t))(-\mu_2\vartheta_b\mathbb{E}\{\int_0^t \|X(s)\|_1 ds | X_c(0), r_0\} - \mu_3\mathbb{E}\{\int_0^t \|L(s)\|_1 ds | L_0, r_0\}).
\end{aligned} \tag{29}$$

Due to $\mathbb{E}\{V(X_c(t), r_t = p)\} > 0$, it is clear that

$$\lim_{t\to\infty} \mathbb{E}\{\int_0^t \|X(s)\|_1 ds | X_c(0), r_0\} \le \sum_{b=1}^m \Theta_b(\phi(t)) \frac{1}{\mu_2 \vartheta_b} V(X_c(0), r_0) < \infty,$$
$$\lim_{t\to\infty} \mathbb{E}\{\int_0^t \|L(s)\|_1 ds | L_0, r_0\} \le \frac{1}{\mu_3} V(X_c(0), r_0) < \infty.$$

Therefore, the systems (5) and (20) are stochastically stable by Definition 2.

Next, consider the case $W(t) \ne 0$. Under the zero initial condition, we have $\mathbb{E}\{V(X_c(t), r_t)\} = \mathbb{E}\{\int_0^t \Gamma V(X_c(t), r_t)\} > 0$. Then,

$$\begin{aligned}
\mathbb{E}\left\{\int_0^t (\|Y(t)\|_1 - \gamma\|W(t)\|_1) dt\right\} \quad &\le \sum_{b,n=1}^m \Theta_b(\phi(t)) \Theta_n(\phi(t)) (\mathbb{E}\{\int_0^t (X^\top(t)(I_N \otimes C_{bp}^\top) \\
&\times \mathbf{1}_{nN} - \gamma W^\top(t) \mathbf{1}_{nN} + X^\top(t)(\mathbf{1}_N \otimes (A_{bp}^\top \eta_p \\
&+ \underline{\alpha} \sum_{\iota=1}^n z_{np}^{(\iota)}) + \epsilon(\mathbf{1}_N \otimes F_{bp}^\top \eta_p) + \sum_{q=1}^J \lambda_{pq}(\mathbf{1}_N \\
&\otimes \eta_q)) + L^\top(t)(\mathbf{1}_N \otimes Q_{bp}^\top \xi_p + \sum_{q=1}^J \lambda_{pq}(\mathbf{1}_N \\
&\otimes \xi_q)) + W^\top(t)(\mathbf{1}_N \otimes E_{bp}^\top \eta_p)) dt\}).
\end{aligned}$$

Using (22d) and (22e) follows that

$$\mathbf{1}_N \otimes (A_{bp}^\top \eta_p + \underline{\alpha} \sum_{\iota=1}^n z_{np}^{(\iota)} + \epsilon(\mathbf{1}_N \otimes F_{bp}^\top \eta_p)) \prec \mathbf{1}_N \otimes (-\mu_2 C_{bp}^\top \mathbf{1}_n)$$

$$\mathbf{1}_N \otimes Q_{bp}^\top \xi_p + \sum_{q=1}^J \lambda_{pq}(\mathbf{1}_N \otimes \xi_q) \prec \mathbf{1}_N \otimes (-\mu_3 \mathbf{1}_N).$$

Then,

$$\begin{aligned}
\mathbb{E}\left\{\int_0^t (\|Y(t)\|_1 - \gamma\|W(t)\|_1) dt\right\} \quad &\le \sum_{b=1}^m \Theta_b(\phi(t)) (\mathbb{E}\{\int_0^t (1 - \mu_2) X^\top(t) \\
&\times (I_N \otimes C_{bp}^\top) \mathbf{1}_{nN} - \mu_3 L^\top(t)(\mathbf{1}_N \otimes \mathbf{1}_N) \quad (30) \\
&+ W^\top(t)(\mathbf{1}_N \otimes (E_{bp}^\top \eta_p - \gamma \mathbf{1}_n)) dt\}).
\end{aligned}$$

By (22f), $I_N \otimes (E_{bp}^\top \eta_p - \gamma\mathbf{1}_n) \prec 0$. Thus, $\mathbb{E}\{\int_0^\infty (\|Y(t)\|_1 - \gamma\|W(t)\|_1) dt\} < 0$. By Definition 2, the systems (5) and (20) are stochastically $L_1$ stable. □

**Remark 5** When analyzing the dynamic behavior of fuzzy PMJCNs, it is crucial to consider the dynamic characteristics of both NSs and LSs. This increases the complexity of the stability analysis. In Theorem 2, the coupling relationship is embodied in the correlation term involving the link state in NSs and the coupling term involving the node state in LSs. However, the coupling term design proposed in Theorem 1 is mainly concerned with the stability of LSs and the coupling term is independent of the node state. This design form allows us to handle link dynamics independently without considering the impact of the node state. This coupling item design is suitable for those cases that the link dynamics has a significant impact on the overall system performance, but the correlation with the node state is not strong. In contrast, the coupling term design in Theorem 2 is more general because it takes into account the effect of node states on the coupling term.

**Remark 6** In this paper, the positivity of the system is taken for any initial time. Since the state equation of NSs contains the state variable of LSs, it is not feasible to prove that the state of the system is non-negative at any time directly by checking the coefficient matrix of the systems. Therefore, an indirect but rigorous method is adopted in this paper. First, it is proved that from any initial time, the coefficient matrix of the differential equation of LSs satisfies

the positive system condition. Thus, the non-negativity of the state at the next time is guaranteed. We can then argue recursively that for all non-negative link initial conditions, the state of LSs will remain non-negative. Finally, it is derived that the state of NSs will also remain non-negative for any non-negative initial conditions. This method not only ensures the rigorism of the demonstration process, but also provides an effective analytical framework for verifying the positivity of complex coupled systems.

## Synchronization

This section will discuss the synchronization of fuzzy PMJCNs consisting of NSs and LSs.

Let $s(t)$ denote the state of an isolated node given by:

**Rule** $b$: IF $\psi_1(t)$ is $\phi_{b1}$, $\psi_2(t)$ is $\phi_{b2}$, $\cdots$, and $\psi_g(t)$ is $\phi_{bg}$, THEN

$$\begin{aligned} \dot{s}(t) &= \widetilde{A}_{bp}s(t) + \widetilde{F}_{bp}f(s(t)) + \widetilde{B}_{bp}\widetilde{u}(t), \\ \widetilde{y}(t) &= \widetilde{C}_{bp}s(t), \end{aligned} \tag{31}$$

where $s(t) \in \mathbb{R}^n, \widetilde{y}(t) \in \mathbb{R}^n, \widetilde{u}(t) \in \mathbb{R}^n$ denote state, output and control input of the isolated node, respectively; $\widetilde{A}_{bp}, \widetilde{B}_{bp}, \widetilde{C}_{bp}, \widetilde{F}_{bp}$ are given matrices of appropriate dimensions, and it is assumed that $\widetilde{A}_{bp}$ is a Metzler matrix, $\widetilde{B}_{bp} \succeq 0, \widetilde{C}_{bp} \succeq 0, \widetilde{F}_{bp} \succeq 0$. By the fuzzy inference method, the defuzzified fuzzy system (31) becomes

$$\begin{aligned} \dot{s}(t) &= \sum_{b=1}^m \Theta_b(\phi(t))(\widetilde{A}_{bp}s(t) + \widetilde{F}_{bp}f(s(t)) + \widetilde{B}_{bp}\widetilde{u}(t)), \\ \widetilde{y}(t) &= \sum_{b=1}^m \Theta_b(\phi(t))\widetilde{C}_{bp}s(t). \end{aligned} \tag{32}$$

Based on the fuzzy model, the controller is designed as:

**Rule** $n$: IF $\psi_1(t)$ is $\phi_{n1}$, $\psi_2(t)$ is $\phi_{n2}$, $\cdots$, and $\psi_g(t)$ is $\phi_{ng}$, THEN

$$\widetilde{u}(t) = \widetilde{K}_{np}s(t),$$

where $\widetilde{K}_{np}$ is the control gain matrix for each $n = 1, 2, \cdots, m$. Then,

$$\widetilde{u}(t) = \sum_{n=1}^m \Theta_n(\phi(t))\widetilde{K}_{np}s(t). \tag{33}$$

Define the synchronization error between the $i$th node and the isolated node: $e_i(t) = x_i(t) - s(t)$. Then,

$$\begin{aligned} \dot{e}_i(t) &= \sum_{b=1}^m \Theta_b(\phi(t))\Big(A_{bp}e_i(t) + F_{bp}\hat{f}(e_i(t)) + \kappa \sum_{j=1}^N l_{ij}(t)\Lambda_{bp}(\hat{h}_j(e_j(t)) + h_j(s(t))) \\ &\quad + B_{bp}u_i(t) + E_{bp}\omega_i(t) + (A_{bp} - \widetilde{A}_{bp} - \widetilde{B}_{bp}\widetilde{K}_{np})s(t) + (F_{bp} - \widetilde{F}_{bp})f(s(t))\Big), \\ \hat{y}_i(t) &= \sum_{b=1}^m \Theta_b(\phi(t))(C_{bp}e_i(t) + (C_{bp} - \widetilde{C}_{bp})s(t)), \end{aligned} \tag{34}$$

where $\hat{f}(e_i(t)) = f(x_i(t)) - f(s(t))$, $\hat{h}_j(e_j(t)) = h(x_j(t)) - h(s(t))$ and $\hat{y}_i(t)$ is the controlled output of the error systems. Based on the fuzzy model, the controller is designed as:

**Rule** $n$: IF $\psi_1(t)$ is $\phi_{n1}$, $\psi_2(t)$ is $\phi_{n2}$, $\cdots$, and $\psi_g(t)$ is $\phi_{ng}$, THEN

$$u_i(t) = K_{np}e_i(t),$$

where $K_{np}$ is the controller gain matrix. Then,

$$u_i(t) = \sum_{n=1}^m \Theta_n(\phi(t))K_{np}e_i(t). \tag{35}$$

Define $e(t) = (e_1^\top(t), e_2^\top(t), ..., e_N^\top(t))^\top$, $\hat{Y}(t) = (\hat{y}_1^\top(t), \hat{y}_2^\top(t), ..., \hat{y}_N^\top(t))^\top$, $S(t) = (s^\top(t), s^\top(t), ..., s^\top(t))^\top$, $F(s(t)) = (f^\top(s(t)), f^\top(s(t)), ..., f^\top(s(t)))^\top$, $F(e(t)) = (\hat{f}^\top(e_1(t)), \hat{f}^\top(e_2(t)), ..., \hat{f}^\top(e_N(t)))^\top$. By substituting (35) into (34) and using the properties of the Kronecker product, the closed-loop system is:

$$
\begin{aligned}
\dot{e}(t) \quad &= \sum_{b,n=1}^m \Theta_b(\phi(t))\Theta_n(\phi(t))\big((I_N \otimes (A_{bp} + B_{bp}K_{np}))e(t) \\
&\quad + (I_N \otimes F_{bp})F(e(t)) + \kappa(I_N \otimes \Lambda_{bp}H(x(t)))L(t) + (I_N \otimes E_{bp})W(t) \\
&\quad + (I_N \otimes (A_{bp} - \widetilde{A}_{bp} - \widetilde{B}_{bp}\widetilde{K}_{np}))S(t) + (I_N \otimes (F_{bp} - \widetilde{F}_{bp}))F(s(t))\big), \\
\hat{Y}(t) \quad &= \sum_{b=1}^m \Theta_b(\phi(t))\big((I_N \otimes C_{bp})e(t) + (I_N \otimes (C_{bp} - \widetilde{C}_{bp}))S(t)\big).
\end{aligned}
\tag{36}
$$

Next, the synchronization problem of sytem (5) is transformed into the stability problem of system (36) by following theorem.

**Theorem 3** If there exist constants $\underline{\alpha} > 0$, $\overline{\alpha} > 0$, $\kappa > 0$, $\mu_0 > 0$, $\mu_1 > 0$, $\mu_2 > 0$, $\mu_3 > 1$, $\gamma > 0$, $\mathbb{R}^n$ vectors $z_{np}^{(\iota)} > 0$, $\widetilde{z}_{np}^{(\iota)} > 0$, $\eta_p > 0$, $\eta_q > 0$, and $\mathbb{R}^N$ vectors $\xi_p > 0$, $\xi_q > 0$ such that the conditions (10), (22c), (22e), (22f), and

$$
\widetilde{A}_{bp} + \varepsilon\widetilde{F}_{bp} + \widetilde{B}_{bp}\sum_{\iota=1}^n \mathbf{1}_n^{(\iota)}\widetilde{z}_{np}^{(\iota)\top} + \mu_0 I_n \succeq 0,
\tag{37a}
$$

$$
A_{bp} - \widetilde{A}_{bp} - \widetilde{B}_{bp}\sum_{\iota=1}^n \mathbf{1}_n^{(\iota)}\widetilde{z}_{np}^{(\iota)\top} + \varepsilon(F_{bp} - \widetilde{F}_{bp}) + \mu_1 I_n \succeq 0,
\tag{37b}
$$

$$
A_{bp} + B_{bp}\sum_{\iota=1}^n \mathbf{1}_n^{(\iota)}z_{np}^{(\iota)\top} + \varepsilon F_{bp} + \mu_2 I_n \succeq 0,
\tag{37c}
$$

$$
\underline{\delta} \leq \mathbf{1}_n^{(\iota)\top}\widetilde{B}_{bp}^\top\eta_p \leq \overline{\delta},
\tag{37d}
$$

$$
A_{bp}^\top\eta_p + \underline{\alpha}\sum_{\iota=1}^n z_{np}^{(\iota)} + \varepsilon F_{bp}^\top\eta_p + \sum_{q=1}^J \lambda_{pq}\eta_q + \mu_3 C_{bp}^\top\mathbf{1}_n \prec 0,
\tag{37e}
$$

$$
A_{bp}^\top\eta_p - \widetilde{A}_{bp}^\top\eta_p - \overline{\delta}\sum_{\iota=1}^l \widetilde{z}_{np}^{(\iota)} + \varepsilon(F_{bp}^\top - \widetilde{F}_{bp}^\top)\eta_p + \mu_3(C_{bp} - \widetilde{C}_{bp})^\top\mathbf{1}_n \prec 0,
\tag{37f}
$$

hold for each $p, q \in \mathfrak{S}$, then the system (5) is positive and reaches the synchronization with $L_1$-gain performance under the coupling term (21) and controllers (33) and (35) with

$$
\widetilde{K}_{np} = \sum_{\iota=1}^n \mathbf{1}_n^{(\iota)}\widetilde{z}_{np}^{(\iota)\top}, \quad K_{np} = \sum_{\iota=1}^n \mathbf{1}_n^{(\iota)}z_{np}^{(\iota)\top}.
\tag{38}
$$

**Proof:** First, the positivity of the system (32) is proved. Give the initial state $s(t_0) \geq 0$. Noting Assumption 1, it follows that $\varepsilon x_{ik}^2(t_0) \leq f_k(x_{ik}(t_0))x_{ik}(t_0) \leq \epsilon x_{ik}^2(t_0)$ by multiplying both sides of the inequality $\varepsilon \leq \frac{f_k(x_i(t_0)) - f_k(s(t_0))}{x_{ik}(t_0) - s_k(t_0)} \leq \epsilon$ by $(x_{ik}(t_0) - s_k(t_0))^2$ and choosing $s_k(t_0) = 0$. Then, $\dot{s}(t_0^+) \geq \sum_{b,n=1}^m \Theta_b(\phi(t_0))\Theta_n(\phi(t_0))(\widetilde{A}_{bp} + \varepsilon\widetilde{F}_{bp} + \widetilde{B}_{bp}\widetilde{K}_{np})s(t_0)$. From (37a) and (38), it derives that $\widetilde{A}_{bp} + \varepsilon\widetilde{F}_{bp} + \widetilde{B}_{bp}\widetilde{K}_{np} + \mu_0 I_n \succeq 0$. By Lemma 1, $\widetilde{A}_{bp} + \varepsilon\widetilde{F}_{bp} + \widetilde{B}_{bp}\widetilde{K}_{np}$ is Metzler. Thus, it holds that $\dot{s}(t_0^+) \geq 0$ for $s(t_0) \geq 0$. Then, $s(t) \geq 0$ for any initial state $s(t_0) \geq 0$ by recursive derivation.

Next, the positivity of the error system (36) is considered. Substituting the formula (35) into (34) yields that

$$
\begin{aligned}
\dot{e}_i(t) \quad &= \sum_{b,n=1}^m \Theta_b(\phi(t))\Theta_n(\phi(t))\big((A_{bp} + B_{bp}K_{np})e_i(t) + F_{bp}\hat{f}(e_i(t)) \\
&\quad + \kappa\sum_{j=1}^N l_{ij}(t)\Lambda_{bp}(\hat{h}_j(e_j(t)) + h_j(s(t))) + B_{bp}u_i(t) + E_{bp}\omega_i(t) \\
&\quad + (A_{bp} - \widetilde{A}_{bp} - \widetilde{B}_{bp}\widetilde{K}_{np})s(t) + (F_{bp} - \widetilde{F}_{bp})f(s(t))\big).
\end{aligned}
$$

Give the initial state $e_i(t_0) \succeq 0$. Using Assumption 1, we can obtain $\varepsilon \leq \frac{f_k(x_i(t_0)) - f_k(s(t_0))}{x_{ik}(t_0) - s_k(t_0)} \leq \epsilon$ and $\rho \leq \frac{h_{ik}(x_i(t_0)) - h_{ik}(s(t_0))}{x_{ik}(t_0) - s_k(t_0)} \leq \varrho$. Consequently, it follows that $\varepsilon e_i(t_0) \preceq \hat{f}(e_i(t_0)) \preceq \epsilon e_i(t_0)$ and $\rho e_i(t_0) \preceq \hat{h}_i(e_i(t_0)) \preceq \varrho e_i(t_0)$. Thus,

$$
\begin{aligned}
\dot{e}_i(t_0) \quad &\geq \sum_{b,n=1}^m \Theta_b(\phi(t_0))\Theta_n(\phi(t_0))\big((A_{bp} + B_{bp}K_{np} + \rho\kappa \sum_{j=1}^N l_{ij}(t_0)\Lambda_{bp} \\
&\quad + \varepsilon F_{bp})e_i(t_0) + E_{bp}\omega_i(t_0) + (A_{bp} - \widetilde{A}_{bp} - \widetilde{B}_{bp}\widetilde{K}_{np} + \varepsilon(F_{bp} - \widetilde{F}_{bp}) \\
&\quad + \rho\kappa \sum_{j=1}^N l_{ij}(t_0)\Lambda_{bp})s(t_0)\big).
\end{aligned}
$$

Let $e(t_0) = (e_1^\top(t_0), e_2^\top(t_0), \ldots, e_N^\top(t_0))^\top$, $S(t_0) = (s^\top(t_0), s^\top(t_0), \ldots, s^\top(t_0))^\top$, and $W(t_0) = (\omega_1^\top(t_0), \omega_2^\top(t_0), \ldots, \omega_N^\top(t_0))^\top$. Then, $\dot{e}(t_0^+) \succeq \sum_{b,n=1}^m \Theta_b(\phi(t_0))\Theta_n(\phi(t_0))(\mathcal{A} \times E(t_0) + \mathcal{B}S(t_0)) + \mathcal{C}W(t_0))$, where

$$
\mathcal{A} = \begin{pmatrix}
\mathbb{K}_{11} & \kappa\rho l_{12}(t_0)\Lambda_{bp} & \cdots & \kappa\rho l_{1N}(t_0)\Lambda_{bp} \\
\kappa\rho l_{21}(t_0)\Lambda_{bp} & \mathbb{K}_{22} & \cdots & \kappa\rho l_{2N}(t_0)\Lambda_{bp} \\
\vdots & \vdots & \ddots & \vdots \\
\kappa\rho l_{N1}(t_0)\Lambda_{bp} & \kappa\rho l_{N2}(t_0)\Lambda_{bp} & \cdots & \mathbb{K}_{NN}
\end{pmatrix},
$$

$\mathbb{K}_{ii} = A_{bp} + B_{bp}K_{np} + \varepsilon F_{bp} + \kappa\rho l_{ii}(t_0)\Lambda_{bp}$, $\mathcal{B} = I_N \otimes (A_{bp} - \widetilde{A}_{bp} - \widetilde{B}_{bp}\widetilde{K}_{np} + \varepsilon(F_{bp} - \widetilde{F}_{bp}) + \rho\kappa \sum_{j=1}^N l_{ij}(t_0)\Lambda_{bp})$, and $\mathcal{C} = I_N \otimes E_{bp}$. From (10) and (37c), it derives that $A_{bp} + B_{bp}K_{np} + \varepsilon F_{bp} + \mu_2 I_n \succeq 0$. By Lemma 1, $A_{bp} + B_{bp}K_{np} + \varepsilon F_{bp}$ is Metzler, that is, $A_{bp} + B_{bp}K_{np} + \varepsilon F_{bp} + \kappa\rho l_{ii}(t_0)\Lambda_{bp}$ is Metzler for $i = \{1, 2, \ldots, N\}$. Noting the condition $l_{ij}(t_0) > 0$, we have that $\kappa\rho l_{ij}(t_0)\Lambda_{bp} \succeq 0$ for $i \neq j$. Then, $\mathcal{A}$ is a Metzler matrix. From (37b) and (38), it holds that $I_N \otimes (A_{bp} - \widetilde{A}_{bp} - \widetilde{B}_{bp}\widetilde{K}_{np} + \varepsilon(F_{bp} - \widetilde{F}_{bp}) + \mu_1 I_n) \succeq 0$. By Lemma 1, $I_N \otimes (A_{bp} - \widetilde{A}_{bp} - \widetilde{B}_{bp}\widetilde{K}_{np} + \varepsilon(F_{bp} - \widetilde{F}_{bp}))$ is a Metzler matrix. Owing to $\rho\kappa \sum_{j=1}^N l_{ij}(t_0)\Lambda_{bp} \succ 0$ and $E_{bp} \succeq 0$, we obtain $\mathcal{B} \succ 0$ and $\mathcal{C} \succeq 0$. Thus, $\dot{e}(t_0^+) \succeq 0$ for $\dot{e}(t_0) \succeq 0$. Then, $e(t) \succeq 0$ for any initial state $e(t_0) \succeq 0$ by recursive derivation. Therefore, the system (36) is positive by Lemma 2.

Choose the stochastic CLF as: $V(e_c(t), r_t) = e_c^\top(t)\varsigma$, where $e_c(t) = (e^\top(t), L^\top(t))^\top$, $\varsigma = (\mathbf{1}_N^\top \otimes \eta_p^\top, \mathbf{1}_N^\top \otimes \xi_p^\top)^\top$, and $\eta_p = M\xi_p$. Then, the weak infinitesimal operator of $V(e_c(t), r_t)$ is:

$$
\begin{aligned}
\Gamma V(e_c(t), r_t) \quad &= \sum_{b,n=1}^m \Theta_b(\phi(t))\Theta_n(\phi(t))(e^\top(t)(\mathbf{1}_N \otimes (A_{bp}^\top + K_{np}^\top B_{bp}^\top)\eta_p \\
&\quad + \sum_{q=1}^J \lambda_{pq}(\mathbf{1}_N \otimes \eta_q)) + F^\top(e(t))(\mathbf{1}_N \otimes F_{bp}^\top \eta_p) \\
&\quad + W^\top(t)(\mathbf{1}_N \otimes E_{bp}^\top \eta_p) + L^\top(t)(\mathbf{1}_N \otimes \kappa H^\top(x(t))\Lambda_{bp}^\top \eta_p) \\
&\quad + S^\top(t)(\mathbf{1}_N \otimes (A_{bp}^\top - \widetilde{A}_{bp}^\top - \widetilde{K}_{np}^\top \widetilde{B}_{bp}^\top)\eta_p) + F^\top(S(t))(\mathbf{1}_N \\
&\quad \otimes (F_{bp}^\top - \widetilde{F}_{bp}^\top)\eta_p) + L^\top(t)(\mathbf{1}_N \otimes (Q_{bp}^\top \xi_p - \kappa H^\top(x(t)) \\
&\quad \times \Lambda_{bp}^\top M\xi_p - \mathbf{1}_N \xi_p^\top \xi_p + \sum_{q=1}^J \lambda_{pq}(\mathbf{1}_N \otimes \xi_q)))).
\end{aligned} \tag{39}
$$

Since $\eta_p = M\xi_p$, it yields that

$$
\begin{aligned}
\Gamma V(e_c(t), r_t) \quad &= \sum_{b,n=1}^m \Theta_b(\phi(t))\Theta_n(\phi(t))(e^\top(t)(\mathbf{1}_N \otimes (A_{bp}^\top + K_{np}^\top B_{bp}^\top)\eta_p \\
&\quad + \sum_{q=1}^J \lambda_{pq}(\mathbf{1}_N \otimes \eta_q)) + F^\top(e(t))(\mathbf{1}_N \otimes F_{bp}^\top \eta_p) \\
&\quad + W^\top(t)(\mathbf{1}_N \otimes E_{bp}^\top \eta_p) + S^\top(t)(\mathbf{1}_N \otimes (A_{bp}^\top - \widetilde{A}_{bp}^\top - \widetilde{K}_{np}^\top \widetilde{B}_{bp}^\top)\eta_p) \\
&\quad + F(S(t))^\top(\mathbf{1}_N \otimes (F_{bp}^\top - \widetilde{F}_{bp}^\top)\eta_p) + L^\top(t)(\mathbf{1}_N \otimes (Q_{bp}^\top \xi_p \\
&\quad - \mathbf{1}_N \xi_p^\top \xi_p + \sum_{q=1}^J \lambda_{pq}(\mathbf{1}_N \otimes \xi_q)))).
\end{aligned} \tag{40}
$$

From Assumption 1, we have $F^\top(e(t)) \preceq \epsilon e^\top(t)$ and $F^\top(S(t)) \preceq \epsilon S^\top(t)$. Then,

$$
\begin{aligned}
\Gamma V(e_c(t), r_t) \quad &\leq \sum_{b,n=1}^{m} \Theta_b(\phi(t)) \Theta_n(\phi(t)) (e^\top(t)(\mathbf{1}_N \otimes (A_{bp}^\top + K_{np}^\top B_{bp}^\top \\
&+ \epsilon F_{bp}^\top) \eta_p + \sum_{q=1}^{J} \lambda_{pq}(\mathbf{1}_N \otimes \eta_q)) + F^\top(e(t))(\mathbf{1}_N \otimes F_{bp}^\top \eta_p) \\
&+ W^\top(t)(\mathbf{1}_N \otimes E_{bp}^\top \eta_p) + S^\top(t)(\mathbf{1}_N \otimes (A_{bp}^\top - \widetilde{A}_{bp}^\top \\
&- \widetilde{K}_{np}^\top \widetilde{B}_{bp}^\top) \eta_p) + \epsilon S^\top(t)(\mathbf{1}_N \otimes (F_{bp}^\top - \widetilde{F}_{bp}^\top) \eta_p) + L^\top(t)(\mathbf{1}_N \\
&\otimes (Q_{bp}^\top \xi_p - \mathbf{1}_N \xi_p^\top \xi_p + \sum_{q=1}^{J} \lambda_{pq}(\mathbf{1}_N \otimes \xi_q)))).
\end{aligned} \tag{41}
$$

From (10), (22c), (37d), and (38), it deduces that

$$
\begin{aligned}
\Gamma V(e_c(t), r_t) \quad &\leq \sum_{b,n=1}^{m} \Theta_b(\phi(t)) \Theta_n(\phi(t))(L^\top(t)(\mathbf{1}_N \otimes Q_{bp}^\top \xi_p + \sum_{q=1}^{J} \lambda_{pq}(\mathbf{1}_N \otimes \xi_q)) \\
&+ e^\top(t)(\mathbf{1}_N \otimes (A_{bp}^\top \eta_p + \underline{\alpha} \sum_{\iota=1}^{n} z_{np}^{(\iota)} + \epsilon F_{bp}^\top \eta_p) + \sum_{q=1}^{J} \lambda_{pq}(\mathbf{1}_N \otimes \eta_q)) \\
&+ S^\top(t)(\mathbf{1}_N \otimes (A_{bp}^\top \eta_p - \widetilde{A}_{bp}^\top \eta_p - \overline{\delta} \sum_{\iota=1}^{l} \widetilde{z}_{np}^{(\iota)} + \epsilon(F_{bp}^\top - \widetilde{F}_{bp}^\top) \eta_p)) \\
&+ W^\top(t)(\mathbf{1}_N \otimes E_{bp}^\top \eta_p)).
\end{aligned}
$$

Consider the case $W(t) = 0$. Combining (22e), (37e) and (37f) yields that

$$
\begin{aligned}
&\mathbf{1}_N \otimes (Q_{bp}^\top \xi_p + \sum_{q=1}^{J} \lambda_{pq} \xi_q) \prec \mathbf{1}_N \otimes (-\mu_3 \mathbf{1}_N), \\
&\mathbf{1}_N \otimes (A_{bp}^\top \eta_p + \underline{\alpha} \sum_{\iota=1}^{n} z_{np}^{(\iota)} + \epsilon F_{bp}^\top \eta_p + \sum_{q=1}^{J} \lambda_{pq} \eta_q) \prec \mathbf{1}_N \otimes (-\mu_3 C_{bp}^\top \mathbf{1}_n), \\
&\mathbf{1}_N \otimes (A_{bp}^\top \eta_p - \widetilde{A}_{bp}^\top \eta_p - \overline{\delta} \sum_{\iota=1}^{l} \widetilde{z}_{np}^{(\iota)} + \epsilon(F_{bp}^\top - \widetilde{F}_{bp}^\top) \eta_p) \prec \mathbf{1}_N \otimes (-\mu_3 (C_{bp} - \widetilde{C}_{bp})^\top \mathbf{1}_n).
\end{aligned}
$$

Then,

$$
\Gamma V(e_c(t), r_t) \quad \leq \sum_{b=1}^{m} \Theta_b(\phi(t))(-\mu_3 \vartheta_b \|e\|_1 - \mu_3 \widetilde{\vartheta}_b \|S\|_1 - \mu_3 \|L\|_1), \tag{42}
$$

where $\widetilde{\vartheta}_b = \min_{p \in \mathfrak{S}} \{ \min_{i=1,2,\dots,n} \Sigma_j^n \widetilde{C}_{bp}^{(ij)} \}$ and $\widetilde{C}_{bp}^{(ij)}$ is the $i$th row $j$th column element of $\widetilde{C}_{bp}$. Applying Dynkin's formula to (42) yields that

$$
\begin{aligned}
&\mathbb{E}\{V(e_c(t), r_t)\} - V(e_c(0), r_0) \\
&\leq \sum_{b=1}^{m} \Theta_b(\phi(t))(-\mu_3(\vartheta_b \mathbb{E}\{\int_0^t \|e(s)\|_1 ds | e(0), r_0\} \\
&- \widetilde{\vartheta}_b \mathbb{E}\{\int_0^t \|S(s)\|_1 ds | S(0), r_0\} - \mathbb{E}\{\int_0^t \|L(s)\|_1 ds | L_0, r_0\})).
\end{aligned} \tag{43}
$$

Due to $\mathbb{E}\{V(e(t), r_t)\} > 0$, we can obtain

$$
\begin{aligned}
&\lim_{t \to \infty} \mathbb{E}\{\int_0^t \|e(s)\|_1 ds | e(0), r_0\} \leq \sum_{b=1}^{m} \Theta_b(\phi(t)) \frac{1}{\mu_3 \vartheta_b} V(e_c(0), r_0) < \infty, \\
&\lim_{t \to \infty} \mathbb{E}\{\int_0^t \|S(s)\|_1 ds | S(0), r_0\} \leq \sum_{b=1}^{m} \Theta_b(\phi(t)) \frac{1}{\mu_3 \widetilde{\vartheta}_b} V(e_c(0), r_0) < \infty, \\
&\lim_{t \to \infty} \mathbb{E}\{\int_0^t \|L(s)\|_1 ds | L_0, r_0\} \leq \frac{1}{\mu_3} V(e_c(0), r_0) < \infty.
\end{aligned}
$$

Therefore, the system (36) is stochastically stable by Definition 2.

Next, consider the case $W(t) \neq 0$. Under the zero initial condition, we have $\mathbb{E}\{V(e_c(t), r_t)\} = \mathbb{E}\{\int_0^t \Gamma V(e_c(t), r_t)\} > 0$. Then,

$$
\begin{aligned}
\mathbb{E}\left\{\int_0^t (\|\hat{Y}(t)\|_1 - \gamma\|W(t)\|_1)dt\right\} &\leq \sum_{b,n=1}^m \Theta_b(\phi(t))\Theta_n(\phi(t))(\mathbb{E}\{\int_0^t (e^\top(t)(I_N \otimes C_{bp}^\top) \\
&\quad \times \mathbf{1}_{nN} + S^\top(t)(I_N \otimes (C_{bp} - \widetilde{C}_{bp})^\top)\mathbf{1}_{nN} \\
&\quad - \gamma W^\top(t)\mathbf{1}_{nN} + e^\top(t)(\mathbf{1}_N \otimes (A_{bp}^\top \eta_p + \underline{\alpha}\sum_{\iota=1}^n z_{np}^{(\iota)}) \\
&\quad + \epsilon(\mathbf{1}_N \otimes F_{bp}^\top \eta_p) + \sum_{q=1}^J \lambda_{pq}(\mathbf{1}_N \otimes \eta_q)) \\
&\quad + W^\top(t)(\mathbf{1}_N \otimes E_{bp}^\top \eta_p) + L^\top(t)(\mathbf{1}_N \otimes Q_{bp}^\top \xi_p \\
&\quad + \sum_{q=1}^J \lambda_{pq}(\mathbf{1}_N \otimes \xi_q)) + S^\top(t)(\mathbf{1}_N \otimes (A_{bp}^\top \eta_p - \widetilde{A}_{bp}^\top \eta_p \\
&\quad - \overline{\delta}\sum_{\iota=1}^l \widetilde{z}_{np}^{(\iota)} + \epsilon(F_{bp}^\top - \widetilde{F}_{bp}^\top)\eta_p))dt\}).
\end{aligned}
$$

By (22e), (37b), and (37f), it gives

$$
\mathbf{1}_N \otimes (Q_{bp}^\top \xi_p + \sum_{q=1}^J \lambda_{pq}\xi_q) \prec \mathbf{1}_N \otimes (-\mu_3 \mathbf{1}_N),
$$

$$
\mathbf{1}_N \otimes (A_{bp}^\top \eta_p + \underline{\alpha}\sum_{\iota=1}^n z_{np}^{(\iota)} + \epsilon F_{bp}^\top \eta_p + \sum_{q=1}^J \lambda_{pq}\eta_q) \prec \mathbf{1}_N \otimes (-\mu_3 C_{bp}^\top \mathbf{1}_n),
$$

$$
\mathbf{1}_N \otimes (A_{bp}^\top \eta_p - \widetilde{A}_{bp}^\top \eta_p - \overline{\delta}\sum_{\iota=1}^l \widetilde{z}_{np}^{(\iota)} + \epsilon(F_{bp}^\top - \widetilde{F}_{bp}^\top)\eta_p) \prec \mathbf{1}_N \otimes (-\mu_3(C_{bp} - \widetilde{C}_{bp}^\top)\mathbf{1}_n).
$$

Then,

$$
\begin{aligned}
\mathbb{E}\left\{\int_0^t (\|\hat{Y}(t)\|_1 - \gamma\|W(t)\|_1)dt\right\} &\leq \sum_{b=1}^m \Theta_b(\phi(t))(\mathbb{E}\{\int_0^t (1 - \mu_3) \\
&\quad \times (e^\top(t)(I_N \otimes C_{bp}^\top)\mathbf{1}_{nN} - \mu_3 L^\top(t)(\mathbf{1}_N \otimes \mathbf{1}_N) \\
&\quad + S^\top(t)(I_N \otimes (C_{bp} - \widetilde{C}_{bp})^\top)\mathbf{1}_{nN}) \\
&\quad + W^\top(t)(\mathbf{1}_N \otimes (E_{bp}^\top \eta_p - \gamma\mathbf{1}_n))dt\}).
\end{aligned} \tag{44}
$$

Using (22f) gives $I_N \otimes (E_{bp}^\top \eta_p - \gamma\mathbf{1}_n) \prec 0$. Since $\mu_3 > 1$, then $\mathbb{E}\{\int_0^\infty (\|\hat{Y}(t)\|_1 - \gamma\|W(t)\|_1)dt\} < 0$. Thus, the system (36) is stochastically $L_1$-gain stable by Definition 2, that is, the system (5) reaches synchronization with $L_1$-gain performance under the controller (35).

Theorem 3 proposes a synchronization framework for fuzzy PMJCNs by using the isolated node (32). In the following corollary, it is stated that the state synchronization of NSs is achieved when the coefficient matrix of the new isolated node system (46) is identical to that in NSs. Let $s(t)$ denote the state of the isolated node given by:

**Rule** $b$: IF $\psi_1(t)$ is $\phi_{b1}$, $\psi_2(t)$ is $\phi_{b2}$, $\cdots$, and $\psi_g(t)$ is $\phi_{bg}$, THEN

$$
\dot{s}(t) = A_{bp}s(t) + F_{bp}f(s(t)). \tag{45}
$$

By the fuzzy inference method, the defuzzified fuzzy system (45) becomes

$$
\dot{s}(t) = \sum_{b=1}^m \Theta_b(\phi(t))(A_{bp}s(t) + F_{bp}f(s(t))). \tag{46}
$$

Define the synchronization error between $i$th node and the isolated node: $e_i(t) = x_i(t) - s(t)$. Then,

$$\dot{e}_i(t) = \sum_{b=1}^{m} \Theta_b(\phi(t))(A_{bp}e_i(t) + F_{bp}\hat{f}(e_i(t)) + B_{bp}u_i(t)$$
$$+ \kappa \sum_{j=1}^{N} l_{ij}(t)\Lambda_{bp}h_j(x_j(t)) + E_{bp}\omega_i(t)),$$
$$\hat{y}_i(t) = \sum_{b=1}^{m} \Theta_b(\phi(t))(C_{bp}e_i(t)),$$
(47)

where $\hat{f}(e_i(t)) = f(x_i(t)) - f(s(t))$. Using the properties of the Kronecker product, the closed-loop system is:

$$\begin{aligned}
\dot{e}(t) &= \sum_{b,n=1}^{m} \Theta_b(\phi(t))\Theta_n(\phi(t))((I_N \otimes (A_{bp} + B_{bp}K_{np}))e(t) \\
&\quad + (I_N \otimes F_{bp})F(e(t)) + \kappa(I_N \otimes \Lambda_{bp}H(x(t)))L(t) + (I_N \otimes E_{bp})W(t)), \\
\hat{Y}(t) &= \sum_{b=1}^{m} \Theta_b(\phi(t))(I_N \otimes C_{bp})e(t).
\end{aligned}$$
(48)

**Corollary 1** If there exist constants $\underline{\alpha} > 0$, $\overline{\alpha} > 0$, $\kappa > 0$, $\mu_1 > 0$, $\mu_2 > 1$, $\mu_3 > 0$, $\gamma > 0$ and $\mathbb{R}^n$ vectors $z_{np}^{(t)} < 0$, $\eta_p > 0$, $\eta_q > 0$, and $\mathbb{R}^N$ vectors $\xi_p > 0$, $\xi_q > 0$ such that the conditions (10), (22b), (22c), (22d), (22e), and (22f) hold for each $p, q \in \mathfrak{S}$, then the system (5) reaches the synchronization with $L_1$-gain performance.

**Proof:** The proof process is similar to Theorem 3 and thus it is omitted.

**Remark 7** The synchronization problem of MJCNs was investigated in [22–25]. Compared with existing synchronization results, the proposed approach considers the dynamics and coupling mechanism between NSs and LSs. In order to achieve the synchronization of NSs, the design of the coupling term needs to take into account the dynamic characteristics of two classes of subsystems. In addition, the final values of the states are finite non-zero values and they are independent of initial conditions.

**Remark 8** This paper focuses on the synchronization issue of PCNs, where the connectivity and the network topology evolve dynamically over time. This characteristic is reflected through the proposed LSs model. The synchronization design of traditional CNs typically employs static topological graphs to represent fixed interconnections among nodes and concentrates on the synchronization of nodal dynamic behaviors. This paper needs to consider both node dynamics and link dynamics. Therefore, the corresponding synchronization problem is more challenging. Furthermore, a core feature of positive complex networks is that the state variables remain non-negative under all non-negative initial conditions. Such a positivity constraint is generally not considered in traditional CNs.

## Illustrative example

In the field of electric energy, an integrated power supply network for generation, storage, transmission and distribution is designed to meet the growing challenge of clean energy. The network consists of several nodes, such as power stations and substations, and transmission lines connecting these nodes. Fig. 1 is an energy storage and transmission unit. Fig. 2 gives the network topology through the layout of nodes and links. The isolated nodes represent an independent power generation facility, while the other nodes are connected to each other by transmission lines, forming a complex network. Fig. 3 shows a network switching electric circuit system with fuzzy Markovian parameters, where the current source $E$ simulates the power

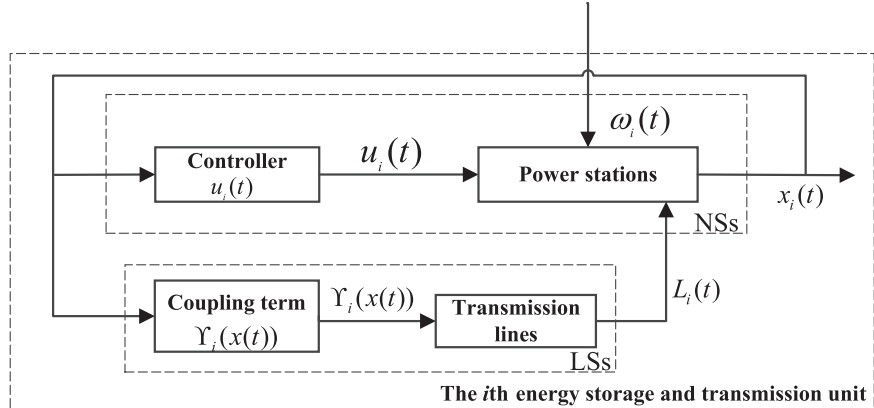

**Fig 1. The *i*th energy storage and transmission unit.**

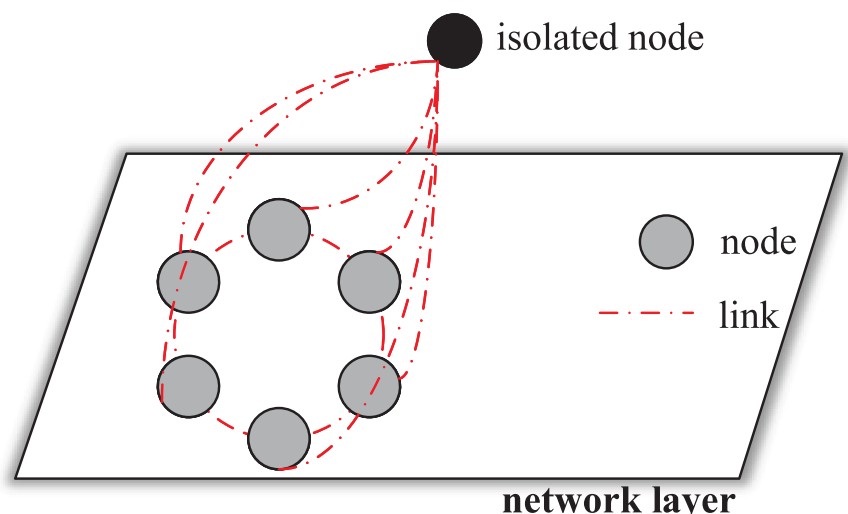

**Fig 2. The network topology.**

output of the generating unit, the capacitors $C_1$ and $C_2$ represent the energy storage elements, and $u_1$ and $u_2$ are the voltages of capacitors with value $C_1$ and $C_2$, respectively. Together with resistors $R_1, R_0, R_2$ and inductors $L$, they form the circuit framework of the $i$th node network. The configuration of these components ensures the positive value of all state variables in the circuit.

According to the Kirvchhoff law, the following equations are obtained:

$$
\begin{aligned}
\frac{du_1}{dt} &= -\frac{R_0 + R_2}{C_1 R'} u_1 + \frac{R_0}{C_1 R'} u_2 + \frac{2R_2(R_0 + R_2)}{C_1 R'} i_L + \frac{R_2}{C_1 R'} E + \frac{R_0 R_2}{C_1 R'} f(u_1), \\
\frac{du_2}{dt} &= \frac{R_0}{C_2 R'} u_1 - \frac{R_0 + R_1}{C_2 R'} u_2 + \frac{R_1 R_0 + R_1 R_2 - R_0 R_2}{C_2 R'} i_L + \frac{R_1}{C_2 R'} E + \frac{R_0 R_1}{C_2 R'} f(u_1), \\
\frac{di_L}{dt} &= \frac{1}{L_p} u_2 - \frac{R_{3p}}{L_p} i_L, \quad p = 1, 2, \cdots, J,
\end{aligned}
\tag{49}
$$

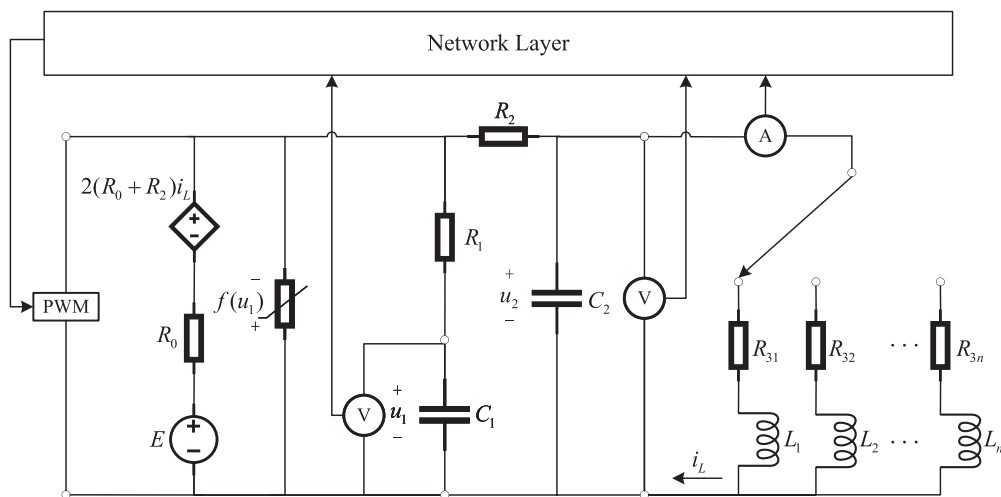

**Fig 3. The *i*th node network circuit framework.**

where $R' = R_1 R_2 + R_0 R_2 + R_0 R_1$, and $f(u_1) = u_1 \sin^2(u_1)$. Let $x_{i1} = u_1$, $x_{i2} = u_2$, $x_{i3} = i_L$, and the external input as $\tilde{w}(t) = E$. The dynamic of $i$th node is described as:

$$\dot{x}_i(t) = A_p x_i(t) + B\tilde{w}(t) + C f(x_{i1}(t)), \tag{50}$$

where

$$A_p = \begin{pmatrix} -\dfrac{R_0 + R_2}{C_1 R'} & \dfrac{R_0}{C_1 R'} & \dfrac{2R_2(R_0 + R_2)}{C_1 R'} \\ \dfrac{R_0}{C_2 R'} & -\dfrac{R_0 + R_1}{C_2 R'} & \dfrac{R_1 R_0 + R_1 R_2 - R_0 R_2}{C_2 R'} \\ 0 & \dfrac{1}{L_p} & -\dfrac{R_{3p}}{L_p} \end{pmatrix},$$

$$B = \begin{pmatrix} \dfrac{R_2}{C_1 R'} \\ \dfrac{R_1}{C_2 R'} \\ 0 \end{pmatrix}, C = \begin{pmatrix} \dfrac{R_0 R_2}{C_1 R'} \\ \dfrac{R_0 R_1}{C_2 R'} \\ 0 \end{pmatrix}.$$

Take $x_{i1}(t)$ as the premise variable, then the normalized membership functions are $\Theta_1(\phi(t)) = \sin^2(x_{i1}(t))$, $\Theta_2(\phi(t)) = 1 - \sin^2(x_{i1}(t))$. The T-S fuzzy model is applied to model the overall network system as follows:

Rule 1: IF $\sin^2(x_{i1}(t)) = 0$, THEN

$$\dot{x}_i(t) = A_{1p} x_i(t) + B\tilde{w}(t) + \kappa \sum_{j=1}^{N} l_{ij}(t) \Lambda_{1p} h_j(x_j(t)),$$

Rule 2: IF $\sin^2(x_{i1}(t)) = 1$, THEN

$$\dot{x}_i(t) = A_{2p} x_i(t) + B\tilde{w}(t) + \kappa \sum_{j=1}^{N} l_{ij}(t) \Lambda_{2p} h_j(x_j(t)),$$

where

$$A_{1p} = \begin{pmatrix} -\dfrac{R_0+R_2}{C_1 R'} & \dfrac{R_0}{C_1 R'} & \dfrac{2R_2(R_0+R_2)}{C_1 R'} \\ \dfrac{R_0}{C_2 R'} & -\dfrac{R_0+R_1}{C_2 R'} & \dfrac{R_1 R_0 + R_1 R_2 - R_0 R_2}{C_2 R'} \\ 0 & \dfrac{1}{L_p} & -\dfrac{R_{3p}}{L_p} \end{pmatrix},$$

$$A_{2p} = \begin{pmatrix} -\dfrac{R_0+R_2+R_0 R_2}{C_1 R'} & \dfrac{R_0}{C_1 R'} & \dfrac{2R_2(R_0+R_2)}{C_1 R'} \\ \dfrac{R_0+R_0 R_1}{C_2 R'} & -\dfrac{R_0+R_1}{C_2 R'} & \dfrac{R_1 R_0 + R_1 R_2 - R_0 R_2}{C_2 R'} \\ 0 & \dfrac{1}{L_p} & -\dfrac{R_{3p}}{L_p} \end{pmatrix},$$

with $\tilde{w}(t) = \Psi_{bp} u_i(t) + \Omega_{bp} f(x_i(t)) + \Xi_{bp} w_i(t)$, and $B_{bp} = B\Psi_{bp}$, $F_{bp} = B\Omega_{bp}$, $E_{bp} = B\Xi_{bp}$, where $\Psi_{bp}$, $\Omega_{bp}$, and $\Xi_{bp}$ are suitable dimension matrices. Choose $R_0 = 1$, $R_1 = 3$, $R_2 = 1$, $R_{31} = 4$, $R_{32} = 5$, $C_1 = 0.6$, $C_2 = 0.6$, $L_1 = 1$, $L_2 = 2$, $B = (0.2381, 0.7143, 0)$, $\Psi_{11} = (0.01, 0.02, 0.03)$, $\Psi_{12} = (0.03, 0.01, 0.01)$, $\Psi_{21} = (0.02, 0.03, 0.01)$, $\Psi_{22} = (0.02, 0.040.01)$, $\Omega_{11} = (0.3, 0.1, 0.5)$, $\Omega_{12} = (0.2, 0.1, 0.3)$, $\Omega_{21} = (0.1, 0.1, 0.4)$, $\Omega_{22} = (0.2, 0.2, 0.3)$, $\Xi_{11} = (0.4, 0.2, 0.5)$, $\Xi_{12} = (0.1, 0.3, 0.2)$, $\Xi_{21} = (0.4, 0.3, 0.1)$, $\Xi_{22} = (0.5, 0.1, 0.3)$. Similarly, the LSs are modeled as $\dot{L}(t) = \sum_{b=1}^{2} \Theta_b(\phi(t))(I_N \otimes Q_{bp})L(t) + \Gamma(x(t))$, where

$$Q_{11} = \begin{pmatrix} -0.81 & 0.14 & 0.12 & 0.13 & 0.16 & 0.23 \\ 0.13 & -0.85 & 0.11 & 0.12 & 0.21 & 0.11 \\ 0.12 & 0.12 & -0.80 & 0.11 & 0.12 & 0.21 \\ 0.11 & 0.14 & 0.12 & -0.89 & 0.16 & 0.13 \\ 0.14 & 0.11 & 0.11 & 0.12 & -0.89 & 0.12 \\ 0.11 & 0.12 & 0.13 & 0.11 & 0.11 & -0.81 \end{pmatrix},$$

$$Q_{12} = \begin{pmatrix} -0.89 & 0.12 & 0.13 & 0.11 & 0.14 & 0.13 \\ 0.11 & -0.85 & 0.12 & 0.13 & 0.18 & 0.12 \\ 0.12 & 0.11 & -0.87 & 0.12 & 0.11 & 0.17 \\ 0.15 & 0.13 & 0.14 & -0.85 & 0.13 & 0.12 \\ 0.13 & 0.12 & 0.12 & 0.14 & -0.81 & 0.14 \\ 0.16 & 0.14 & 0.11 & 0.13 & 0.12 & -0.70 \end{pmatrix},$$

$$Q_{21} = \begin{pmatrix} -0.88 & 0.11 & 0.14 & 0.11 & 0.11 & 0.13 \\ 0.14 & -0.86 & 0.12 & 0.14 & 0.17 & 0.18 \\ 0.11 & 0.14 & -0.80 & 0.12 & 0.16 & 0.16 \\ 0.13 & 0.12 & 0.13 & -0.87 & 0.13 & 0.14 \\ 0.16 & 0.14 & 0.15 & 0.13 & -0.83 & 0.15 \\ 0.12 & 0.13 & 0.11 & 0.15 & 0.12 & -0.79 \end{pmatrix},$$

$$Q_{22} = \begin{pmatrix} -0.83 & 0.11 & 0.12 & 0.16 & 0.13 & 0.21 \\ 0.12 & -0.84 & 0.15 & 0.13 & 0.17 & 0.15 \\ 0.11 & 0.14 & -0.85 & 0.11 & 0.14 & 0.14 \\ 0.15 & 0.16 & 0.13 & -0.81 & 0.16 & 0.11 \\ 0.15 & 0.14 & 0.16 & 0.14 & -0.86 & 0.16 \\ 0.17 & 0.15 & 0.14 & 0.13 & 0.13 & -0.83 \end{pmatrix},$$

$f_i(x_i(t)) = 0.1x_i(t) + \frac{x_i(t)}{x_i^2(t)+1}$ and $h_i(x_i(t)) = (\cos^2(x_{i2}(t)x_{i3}(t)), \sin^2(x_{i1}(t)x_{i3}(t)), \tanh^2$

$(x_{i1}(t)x_{i2}(t)))^\top$. The coupling strength is: $\kappa = 1$. The transition rate matrix is: $\begin{pmatrix} -0.3 & 0.3 \\ 0.2 & -0.2 \end{pmatrix}$.

The inner coupling matrix is chosen as $\Lambda_{11} = diag(0.4, 0.3, 0.2)$, $\Lambda_{12} = diag(0.5, 0.7, 0.4)$, $\Lambda_{21} = diag(0.3, 0.6, 0.4)$, $\Lambda_{22} = diag(0.1, 0.3, 0.6)$. The initial values are: $x(0) = randn(3,1)$ and $L(0) = randn(6,1)$. Give $\overline{\beta} = 1.6$, $\underline{\beta} = 0.4$, $\varepsilon = 0.3$ and $\epsilon = 0.5$. By Theorem 2, we obtain the control protocol gain matrices:

$$K_{11} = \begin{pmatrix} -26.4946 & -13.7489 & -27.1513 \\ -3.0568 & -19.5003 & -15.6439 \\ -1.4405 & -11.3243 & -20.4468 \end{pmatrix}, K_{12} = \begin{pmatrix} -3.6534 & -14.9813 & -19.2998 \\ -10.7956 & -3.8756 & -16.8544 \\ -7.3354 & -41.3567 & -10.2200 \end{pmatrix},$$

$$K_{21} = \begin{pmatrix} -23.6825 & -31.9320 & -13.8775 \\ -1.3961 & -6.1684 & -29.9717 \\ -2.6498 & -12.8818 & -15.5571 \end{pmatrix}, K_{22} = \begin{pmatrix} -2.2580 & -17.4843 & -15.7693 \\ -12.2696 & -9.2145 & -13.7044 \\ -16.4248 & -17.6377 & -39.7504 \end{pmatrix}.$$

Figs 4 and 5 provide the simulations of the state trajectories $x_i(t)$ and $L_i(t)$, $i = 1, 2, 3, 4, 5, 6$, respectively. It can be seen from Figs 4 and 5 that the state trajectories of nodes and links are non-negative respectively and converge asymptotically to zero. The smoothness of convergence indicates that the controller has good performance in terms of stability and response time.

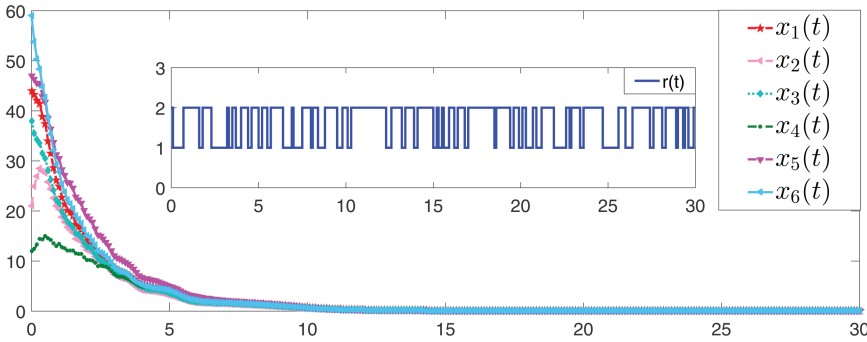

**Fig 4. The trajectories of the states of NSs with the controller (4).**

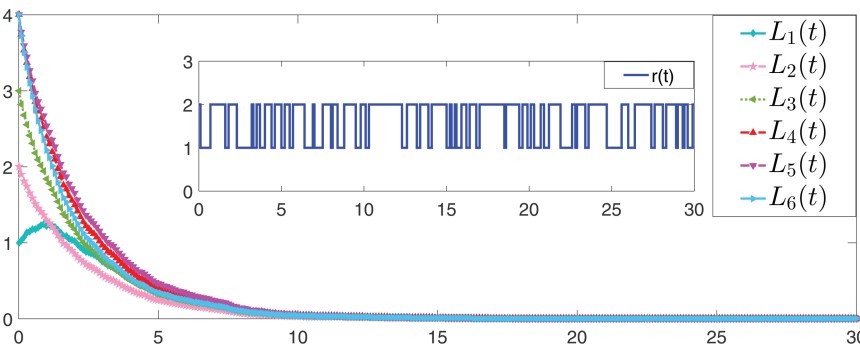

**Fig 5. The trajectories of the states of LSs with the coupling term (21).**

Furthermore, the initial conditions for the isolated node and error are chosen as: $s(0) = randn(3,1)$ and $e(0) = x(0) - s(0)$. By Corollary 1, we obtain the control protocol gain matrices:

$$K_{11} = \begin{pmatrix} -23.6840 & 12.6330 & -25.6052 \\ -3.7964 & -17.2268 & -15.3550 \\ -1.9755 & -11.4113 & -19.6389 \end{pmatrix}, K_{12} = \begin{pmatrix} -5.3107 & -8.6797 & -18.4612 \\ -8.2772 & -2.6338 & -16.3188 \\ -7.1869 & -52.3269 & -11.1626 \end{pmatrix},$$

$$K_{21} = \begin{pmatrix} -19.2124 & -32.1002 - 14.3336 \\ -2.2058 & -2.7781 - 30.4840 \\ -3.3583 & -11.8789 - 15.5454 \end{pmatrix}, K_{22} = \begin{pmatrix} -2.6647 & -17.0960 & -15.3457 \\ -12.8992 & -6.9761 & -13.5901 \\ -13.5668 & -17.8763 & -37.6156 \end{pmatrix}.$$

Fig. 6 gives the simulations of the state trajectories $x_i(t)$ and $s(t), i = 1, 2, 3, 4, 5, 6$, Fig. 7 is the states of LSs, and Fig. 8 shows the tracking error of the six following nodes with respect to the leader $s(t)$ under the action of the controller (35). It can be observed from Fig. 8 that the tracking error of each node gradually approaches zero, which indicates that the synchronization controller has achieved the synchronization performance.

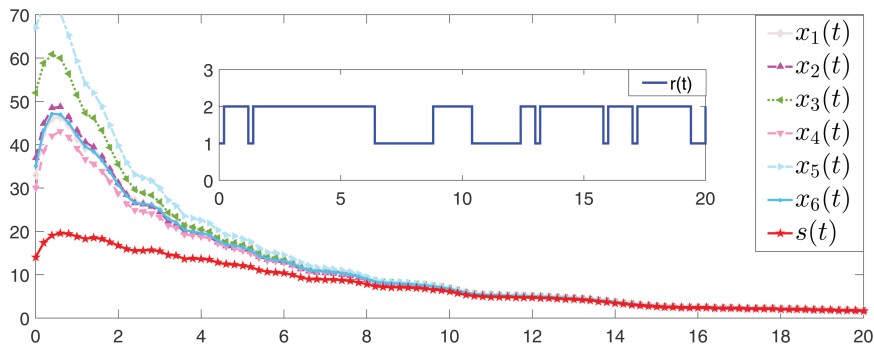

**Fig 6. The state trajectories of NSs.**

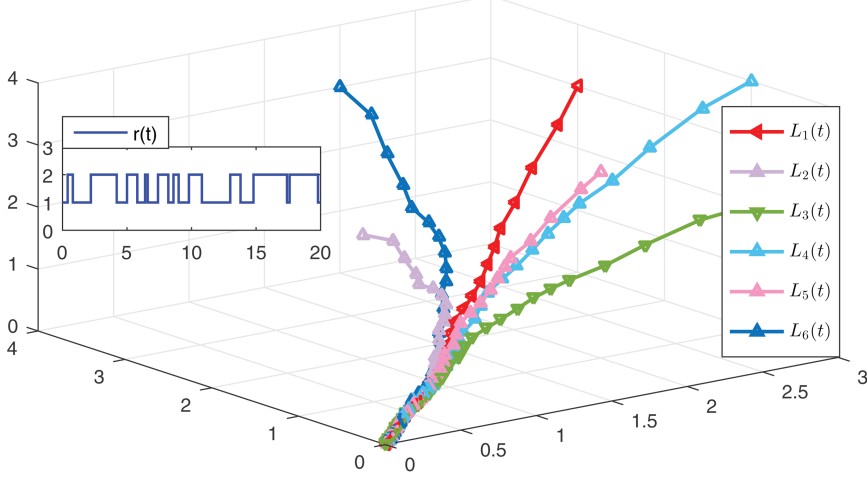

**Fig 7. The state trajectories of LSs.**

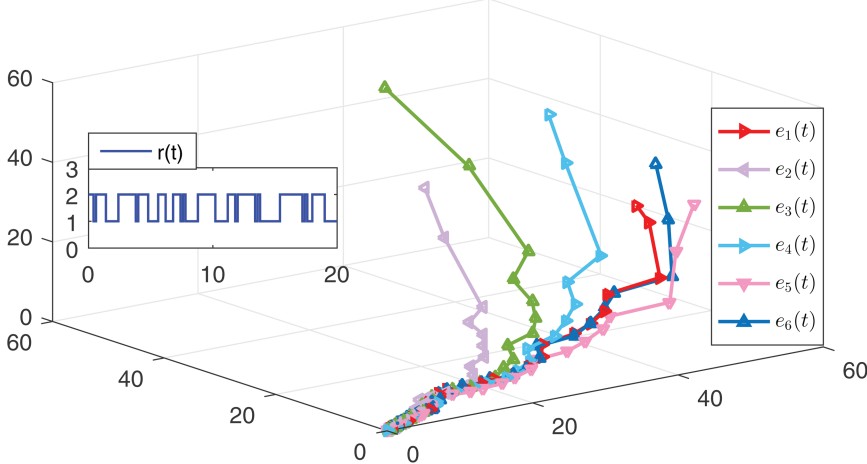

**Fig 8. The trajectories of the error $e(t)$ with the controller (35).**

## Conclusions

This paper constructs a novel fuzzy PMJCNs model, which describes the dynamic behavior of the network through the mutual coupling of NSs and LSs. An innovative coupling mechanism is designed. This mechanism aims to achieve the stability and synchronization performance of the PMJCNs by introducing coupling terms and related controllers. In addition, a LP method is proposed to solve all conditions. Future research will focus on further optimizing the coupling mechanism to improve the stability and synchronization performance of the PMJCNs.

## Author contributions

**Conceptualization:** Junfeng Zhang, Gang Zheng, Bhatti Uzair Aslam.

**Formal analysis:** Junfeng Zhang, Gang Zheng.

**Investigation:** Shouting Hong, Junfeng Zhang.

**Methodology:** Shouting Hong, Junfeng Zhang, Gang Zheng, Bhatti Uzair Aslam.

**Software:** Shouting Hong, Haoyue Yang.

**Supervision:** Junfeng Zhang, Bhatti Uzair Aslam.

**Validation:** Junfeng Zhang, Haoyue Yang.

**Writing – original draft:** Shouting Hong.

**Writing – review & editing:** Junfeng Zhang, Gang Zheng, Haoyue Yang, Bhatti Uzair Aslam.

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
