## [Decision Letter · Decision Letter 0]

14 Feb 2025

PONE-D-25-00197Linear Programming-Based Stabilization and Synchronization of Positive Complex Networks with Dynamic Link SubsystemPLOS ONE

Dear Dr. Zhang,

Thank you for submitting your manuscript to PLOS ONE. After careful consideration, we feel that it has merit but does not fully meet PLOS ONE’s publication criteria as it currently stands. Therefore, we invite you to submit a revised version of the manuscript that addresses the points raised during the review process.

**ACADEMIC EDITOR: **Two reviews have been received for the submission, and both reviewers express a positive assessment of the work. However, they also provide several suggestions for improving its quality. These suggestions include: 1) clarifying the key differences between the synchronization problem discussed in this paper and those in traditional complex dynamic networks; 2) providing a more detailed explanation of the positive proof; and 3) providing a thorough description of the simulation results. The authors are encouraged to address these points.

We look forward to receiving your revised manuscript.

Kind regards,

Gang Wang

Academic Editor

PLOS ONE

3. This research was supported by the National Natural Science Foundation of China (62463007, 62463005, and 62073111), Hainan Province Science and Technology Special Fund (ZDYF2024GXJS003), Science Research Funding of Hainan University (KYQD(ZR)22180), and Postgraduate Innovative Research Funding of Hainan Province (Qhys2023-278, Qhys2023-279 and Qhys2023-280).

Additional Editor Comments:

AE Report: Two reviews have been received for the submission, and both reviewers express a positive assessment of the work. However, they also provide several suggestions for improving its quality. These suggestions include: 1) clarifying the key differences between the synchronization problem discussed in this paper and those in traditional complex dynamic networks; 2) providing a more detailed explanation of the positive proof; and 3) providing a thorough description of the simulation results. The authors are encouraged to address these points.

Reviewers' comments:

Reviewer's Responses to Questions

**Comments to the Author**

1. Is the manuscript technically sound, and do the data support the conclusions?

Reviewer #1: Yes

Reviewer #2: Yes

2. Has the statistical analysis been performed appropriately and rigorously? 

Reviewer #1: Yes

Reviewer #2: Yes

3. Have the authors made all data underlying the findings in their manuscript fully available?

Reviewer #1: Yes

Reviewer #2: Yes

4. Is the manuscript presented in an intelligible fashion and written in standard English?

Reviewer #1: Yes

Reviewer #2: Yes

5. Review Comments to the Author

Reviewer #1: This paper presents a novel model for positive complex networks that integrates dynamic link subsystems to achieve stability and synchronization through innovative coupling mechanisms and control strategies. In my opinion, this is the first work on the dynamic link synchronization control of positive complex networks. This paper contains novelties and has potential applications in other issues of positive complex networks. Some revisions are necessary before acceptance.

1. In the introduction, individual places can be further refined to improve the reading experience of the paper.

2. Some missing symbols in the manuscript. Please check carefully.

3. In Theorem 1, "hold for each i, j" is inconsistent with "hold for each p, q" in Theorem 2 and Theorem 3. Please check and correct.

4. The equation (35) provides a design for a controller. Does it have any impact on synchronization or on Theorem 3? Some statements could be made about it.

5. It would be beneficial if the author could offer a more detailed description of the simulation results.

Reviewer #2: This paper provides a new perspective on the field by constructing a positive complex network model that includes dynamically linked subsystems. An innovative coupling term is designed and a related controller is proposed to achieve stability and synchronization of the network. Through the application of linear programming and copositive Lyapunov functions, this paper provides a feasible design, analysis, and calculation method, which is relatively new in the study of positive complex networks. Although the contributions of this paper are significant, authors are advised to consider the following improvements before final acceptance for publication:

1. The author should proofread the manuscript thoroughly to ensure that all mathematical symbols and formulas have been correctly presented.

2. In the preliminaries, the description of the set is not clear, please check it.

3. There are two different design forms of coupling terms involved in this paper. Please further explain the differences between the two design forms.

4. The arguments about the positivity of the system in this paper all start from a specific moment . I suggest the author providing a more detailed explanation of the positive proof starting from time .

5. What are the fundamental differences between the synchronization problem of positive complex networks discussed in this paper and that of traditional complex dynamic networks?

6. PLOS authors have the option to publish the peer review history of their article (what does this mean?). If published, this will include your full peer review and any attached files.

Reviewer #1: No

Reviewer #2: No

---

## [Author Response · Author response to Decision Letter 1]

24 Feb 2025

Please find the response attached.

---

## [Decision Letter · Decision Letter 1]

10 Mar 2025

Linear Programming-Based Stabilization and Synchronization of Positive Complex Networks with Dynamic Link Subsystem

PONE-D-25-00197R1

Dear Dr. Zhang,

We’re pleased to inform you that your manuscript has been judged scientifically suitable for publication and will be formally accepted for publication once it meets all outstanding technical requirements.

Kind regards,

Gang Wang

Academic Editor

PLOS ONE

Additional Editor Comments (optional):

Reviewers' comments:

Reviewer's Responses to Questions

**Comments to the Author**

1. If the authors have adequately addressed your comments raised in a previous round of review and you feel that this manuscript is now acceptable for publication, you may indicate that here to bypass the “Comments to the Author” section, enter your conflict of interest statement in the “Confidential to Editor” section, and submit your "Accept" recommendation.

Reviewer #1: All comments have been addressed

2. Is the manuscript technically sound, and do the data support the conclusions?

Reviewer #1: Yes

3. Has the statistical analysis been performed appropriately and rigorously? 

Reviewer #1: Yes

4. Have the authors made all data underlying the findings in their manuscript fully available?

Reviewer #1: Yes

5. Is the manuscript presented in an intelligible fashion and written in standard English?

Reviewer #1: Yes

6. Review Comments to the Author

Reviewer #1: (No Response)

7. PLOS authors have the option to publish the peer review history of their article (what does this mean?). If published, this will include your full peer review and any attached files.

Reviewer #1: No

---

## [Editor Report · Acceptance letter]

PONE-D-25-00197R1

PLOS ONE

Dear Dr. Zhang,

I'm pleased to inform you that your manuscript has been deemed suitable for publication in PLOS ONE. Congratulations! Your manuscript is now being handed over to our production team.

Kind regards,

on behalf of

Dr. Gang Wang

Academic Editor

PLOS ONE